# CODEBOOK FEATURES: SPARSE AND DISCRETE INTERPRETABILITY FOR NEURAL NETWORKS

## ABSTRACT

Understanding neural networks is challenging in part because of the dense, continuous nature of their hidden states. We explore whether we can train neural networks to have hidden states that are sparse, discrete, and more interpretable by quantizing their continuous features into what we call **codebook features**. Codebook features are produced by finetuning neural networks with vector quantization bottlenecks at each layer, producing a network whose hidden features are the sum of a small number of discrete vector *codes* chosen from a larger codebook. Surprisingly, we find that neural networks can operate under this extreme bottleneck with only modest degradation in performance. This sparse, discrete bottleneck also provides an intuitive way of **controlling** neural network behavior: first, find codes that activate when the desired behavior is present, then activate those same codes during generation to elicit that behavior. We validate our approach by training codebook Transformers on several different datasets. First, we explore a finite state machine dataset with far more hidden states than neurons. In this setting, our approach overcomes the *superposition* problem by assigning states to distinct codes, and we find that we can make the neural network behave as if it is in a different state by activating the code for that state. Second, we train Transformer language models with up to 410M parameters on two natural language datasets. We identify codes in these models representing diverse, disentangled concepts (ranging from negative emotions to months of the year) and find that we can guide the model to generate different topics by activating the appropriate codes during inference. Overall, codebook features appear to be a promising *unit of analysis and control* for neural networks and interpretability. Our codebase and models are open-sourced.

## 1 INTRODUCTION

The strength of neural networks lies in their ability to learn *emergent* solutions that we could not program ourselves. Unfortunately, the learned programs inside neural networks are challenging to make sense of, in part because they differ from traditional software in important ways. Most strikingly, the *state* of a neural network program, including intermediate computations and features, is implemented in dense, continuous vectors inside of a network. As a result, many different pieces of information are commingled inside of these vectors, violating the software engineering principle of *separation of concerns* (Dijkstra, 1982). Moreover, the continuous nature of these vectors means no feature is ever truly *off* inside of a network; instead, they are activated to varying degrees, vastly increasing the complexity of this state and the possible interactions within it.

A natural question is whether it is possible to recover some of the sparsity and discreteness properties of traditional software systems while preserving the expressivity and learnability of neural networks. To make progress here, we introduce a *structural constraint* into training that *refactors* a network to adhere more closely to these design principles. Specifically, we finetune a network with trainable vector quantization bottlenecks (Gray, 1984) at each layer, which are sparse and discrete. We refer to each vector in this bottleneck as a *code* and the entire library of codes as the *codebook*. See Figure 1 for a visual depiction of this motivation.

The resulting codebooks learned through this process are a promising interface for understanding and controlling neural networks. For example, when we train a codebook language model on the

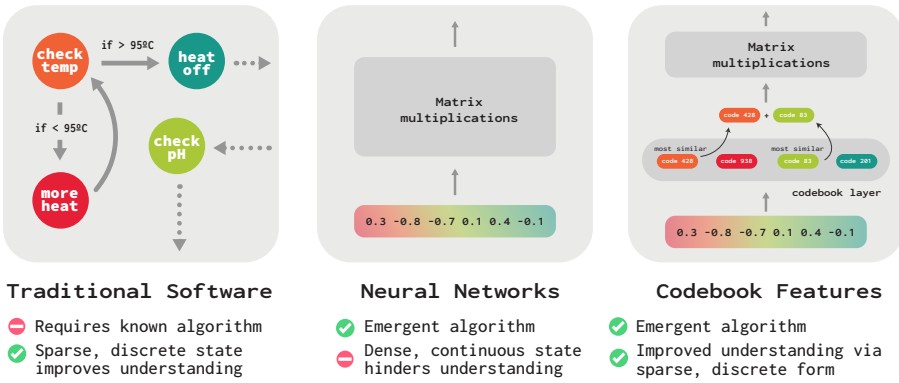

Figure 1: Codebook features attempt to combine the expressivity of neural networks with the sparse, discrete state often found in traditional software.

outputs of a finite state machine, we find a precise mapping between activated codes in different layers of the model to the states of the state machine, overcoming the challenge of *superposition* (Elhage et al., 2022b). Furthermore, we demonstrate a **causal** role for these codes: changing which code is activated during the forward pass causes the network to behave as if it were in a different state. Additionally, we apply codebook features to transformer language models with up to 410M parameters, showing that despite this bottleneck, they can be trained with only modest accuracy degradation compared to the original model. We find codes that activate on a wide range of concepts, spanning punctuation, syntax, lexical semantics, and high-level topics. We then show how to use codebook features to control the topic of a model's generations, providing a practical example of how to use our method to understand and control real language models.

## 2 METHOD

Codebook features aim to improve our understanding and control of neural networks by compressing their activation space with a sparse, discrete bottleneck. Specifically, we aim to learn a set of *discrete states* the network can occupy, of which very few are active during any single forward pass. As we will show later in the paper (Sections 3 and 4), this bottleneck encourages the network to store useful and disentangled concepts in each code. Even more importantly, we show that these interpretations enable us to make causal interventions on the network internals, producing the expected change in the network's behavior. Crucially, codebooks are *learned*, not hand-specified, enabling them to capture behaviors potentially unknown by human researchers.

Concretely, codebook features are produced by replacing a hidden layer's activations with a sparse combination of code vectors. Let $a \in \mathbb{R}^N$ be the activation vector of a given N-dimensional layer in a network. We have a codebook $\mathcal{C} = \{c_1, c_2, ..., c_C\} \in \mathbb{R}^{C \times N}$, where $C$ is the codebook size. To apply the codebook, we first compute the cosine similarities $\text{sim}(a, c_i) = \frac{a \cdot c_i}{|a||c_i|}$ between $a$ and each code vector $c_i$. We then replace $a$ with $\sum_{i \in S} c_i$, where $S$ contains the indices of the top $k$ most similar code vectors. In other words, we activate and sum the $k$ code vectors most similar to the original activation $a$. The value of $k$ controls the bottleneck's sparsity; we aim to make $k$ as small as possible while achieving adequate performance. $k$ is a small fraction of $C$ in our experiments, typically less than 1%, and as a result, we find that codebooks are tight information bottlenecks, transmitting much less information than even 4-bit quantized activations (Appendix B).

While codebook features can be applied to any neural network, we primarily focus on Transformer networks, placing codebooks after either the network's MLP blocks or attention heads. Figure 2 shows the precise location of the codebook for each type of sublayer. Note that this positioning of the codebooks preserves the integrity of the residual stream of the network, which is important for optimizing deep networks (He et al., 2016; Elhage et al., 2021).

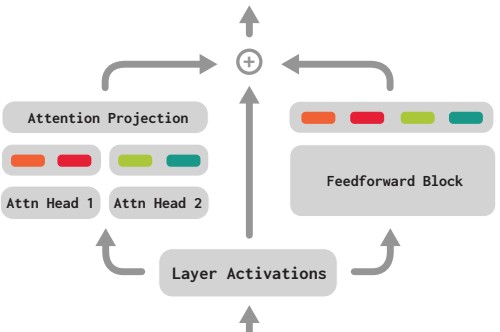

Figure 2: **Applying codebook features to transformers.** *Attention heads*: We add one codebook (depicted by the colored rectangles) for each attention head. The codebook is inserted before the projection into the residual stream. *Feedforward block*: We insert the codebook after the feedforward block, before addition into the residual stream.

## 2.1 TRAINING WITH CODEBOOKS

To obtain codebook features, we add the codebook bottlenecks to existing pretrained models and finetune the model with the original training loss. Thus, the network must learn to perform the task well while adjusting to the discrete codebook bottleneck. Using a pretrained model enables us to produce codebook features more cheaply than training a network from scratch. When finetuning, we use a linear combination of two losses:

**Original training loss**   In our work, we apply codebooks to Transformer-based causal language models and thus use the typical cross-entropy loss these models were trained with: $\mathcal{L}_{\text{LM}}(\theta) = -\sum_{i=1}^{N} \log p_\theta(x_i|x_{<i})$ where $\theta$ represents the model parameters, $x_i$ is the next token of input sequence $x_{<i}$, $p_\theta(x_i|x_{<i})$ is the model's predicted probability of token $x_i$ given input $x_{<i}$, and $N$ is the length of the input sequence.

**Reconstruction loss**   Because we compute the similarity between activations and codebook features using the cosine similarity, which is invariant to magnitude, the code vectors can often grow in size throughout training, leading to instability. For this reason, we find it helpful to add an auxiliary loss to the codes: $\mathcal{L}_{\text{MSE}} = \text{MSE}(\mathcal{C}(a), \text{stop-gradient}(a))$, where $a$ are the input activations to the codebook, $\mathcal{C}(a)$ is its output, and MSE is the mean squared error, to keep the distance between inputs and chosen codes small. The stop gradient means the gradient of this operation only passes through the codebook, not the input $a$, which we found was important to avoid damaging the network's capabilities.[1]

**Final loss and optimization**   The final loss is simply a combination of both losses above $\mathcal{L} = \mathcal{L}_{\text{LM}} + \lambda L_{\text{MSE}}$ where $\lambda$ is a tradeoff coefficient. We set $\lambda$ to 1 in this work. To optimize the codebooks despite the discrete choice of codes, we use the straight-through estimator: we propagate gradients to the codes that were chosen on each forward pass and pass no gradients to the remaining codes (Bengio et al., 2013; van den Oord et al., 2017). We use this strategy to successfully perform end-to-end training of networks up to 24 layers deep, with each layer having a codebook. We defer additional details to Appendix A.

## 2.2 USING CODEBOOKS FOR UNDERSTANDING AND CONTROL

A trained codebook model enables a simple and intuitive way of controlling the network's behavior. This method consists of two phases:

**1) Generating hypotheses for the role of codes.**   Most codes are activated infrequently in the training dataset. We can gain an intuition for the *functional role* of each code in the network's hidden state by retrieving many examples in the dataset where that code was activated. For example, if a code activates mainly around words like "candle," "matches," and "lighters," we might hypothesize that the token is involved in representations of fire. The discrete on-or-off nature of codes makes this task more manageable than looking at continuous values like neuron activations, as past work has speculated that lower-activating neurons can "smuggle" important information across layers, even if many neurons appear interpretable (Elhage et al., 2022a). As we will show in the following

---

[1]We performed preliminary experiments that only used the reconstruction loss (keeping the language model's parameters fixed), similar to a VQ-VAE (van den Oord et al., 2017) at every layer. However, we achieved significantly worse performance. See Table 8 for more details.

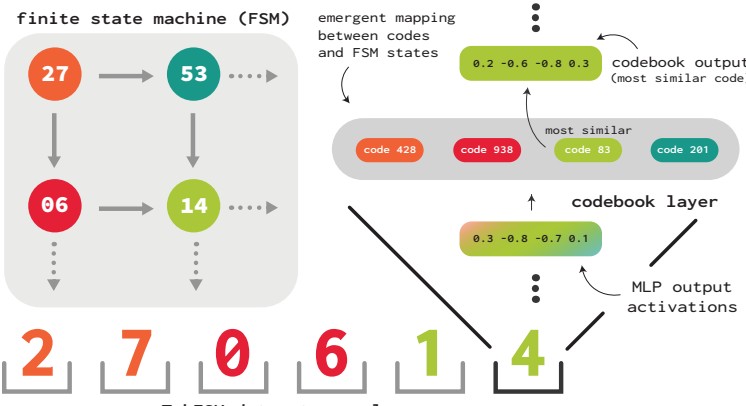

Figure 3: **Codebook features learn the hidden structure of an algorithmic sequence modeling task.** The codebook transformer learns to detect the states of a finite state machine and assigns a code to each state. We can then manipulate these codes to cause the network to make predictions as if it were in a different state.

sections, the codes we discover activate more often on a single interpretable feature, while neurons may activate on many unrelated features. Appendix E.1 discusses the advantages and tradeoffs of codebooks over neuron- and feature direction–based approaches in more detail.

**2) Steering the network by activating codes.** After we have identified codes that reliably activate on the concept we are interested in, we can directly activate those codes to influence the network's behavior. For example, if we identified several codes related to fire, we could activate those codes during generation to produce outputs about fire (e.g., as in Section 4.1). This intervention confirms that the codes have a *causal role* in the network's behavior.

In the following sections, we apply this same two-step procedure across several different datasets, showing that we can successfully gain insight into the network and control its behavior in each case.

## 3 ALGORITHMIC SEQUENCE MODELING

The first setting we consider is an algorithmic sequence modeling dataset called TokFSM. The purpose of this dataset is to create a controlled setting exhibiting some of the complexities of language modeling, *but where the latent features present in the sequence are known*. This setting enables us to evaluate how well the model learns codes that activate on these distinct features. An overview of the section and our findings is shown in Figure 3. Below, we describe the dataset, and then (following Section 2.2) we first generate hypotheses for the role of codes, then show how one can predictably influence the network's behavior by manipulating these codes.

**The TokFSM Dataset** The TokFSM dataset is produced by first constructing a simplified finite state machine (FSM). Our FSM is defined by $(V, E)$ where $V = \{0, \cdots, N - 1\}$ is a set of nodes and $E \subseteq V \times V$ indicates the set of valid transitions from one state to the next. In our setting, we choose $N = 100$ and give each node 10 randomly chosen outbound neighbors, each assigned an equal transition probability $(0.1)$. Entries in the dataset are randomly sampled rollouts of the FSM up to 64 transitions. We tokenize the sequences at the digit level; this gives a sequence length of 128 for each input. For example, if our sampled rollout is [18, 00, 39], we would tokenize it as [1, 8, 0, 0, 3, 9] for the neural network. Thus, the model must learn to detokenize the input into its constituent states, predict the next FSM state, and then retokenize the state to predict the next token.

**Training and evaluating the codebook models** We train 4-layer Transformers with 4 attention heads and an embedding size of 128 based on the GPTNeoX architecture (Black et al., 2022) on the TokFSM dataset. We train several models with different numbers of codes and sparsity values $k$, with codebooks either at the network's attention heads or both the attention heads and MLP Layers (see Figure 2). In Table 1, we report the accuracy of the resulting models both in terms of their language modeling loss, next token accuracy, and their ability to produce valid transitions of the FSM across a generated sequence. The $k = 1$ model with codebooks at only the attention layers achieves comparable performance across all metrics to the original model. At the same time, larger values of $k$ enable the model with codebooks at both attention and MLP blocks to attain comparable performance. It is striking that networks can perform so well despite this extreme bottleneck at every layer. We defer additional training details to Appendix C.1 and ablation studies to Table 8.

| Codebook Type | Loss | LM Acc | State Acc |
|---|---|---|---|
| No Codebook | 1.179 | 46.36 | 96.77 |
| Attn Only $_{k=1,\ C=2k}$ | 1.18 | 46.33 | 96.39 |
| †Attn+MLP $_{k=1,\ C=10k}$ | 1.269 | 45.27 | 63.65 |
| Attn+MLP $_{k=1,\ C=20k}$ | 1.254 | 45.56 | 63.81 |
| Attn+MLP $_{k=4,\ C=20k}$ | 1.192 | 46.20 | 80.69 |
| Attn+MLP $_{k=16,\ C=20k}$ | 1.183 | 46.32 | 91.53 |
| Attn+MLP $_{k=128,\ C=20k}$ | 1.178 | 46.38 | 95.82 |

Table 1: **Performance of original and codebook models on TokFSM.** A $k = 1$ codebook model on only attention layers attains similar performance to the original model, while attention-and-MLP codebooks require a higher $k$ and codebook size $C$ to match performance. † indicates the model we analyze in the rest of the section.

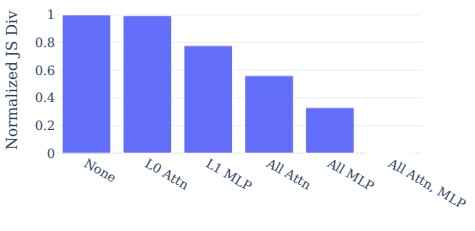

(a) State code interventions

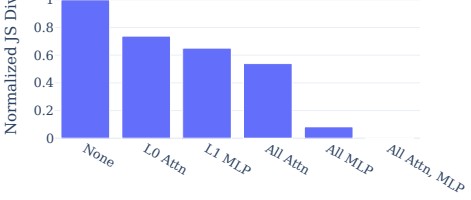

(b) State-plus-digit code interventions

Figure 4: **Interventions on the state and state-plus-digit codes in a sequence.** Changing just the MLP codes to codes associated with another state shifts the output distribution almost entirely to the target state. Changing codes in other layers has a much smaller effect. Normalized JS Div stands for the normalized Jensen-Shannon Divergence, where the initial difference (None) is normalized to 1.

## 3.1 GENERATING HYPOTHESES FOR THE ROLE OF CODES

After training these models, we examine the $k = 1$ attention and MLP codebook transformer following Section 2.2. Looking at activating tokens reveals a wide range of interesting-looking codes. We provide descriptions of these codes along with a table of examples in Table 6, and focus our analysis on two families of codes here: in the last three MLP layers (layers 1, 2, and 3), we identify **state codes** that reliably activate on the second token of a specific state (of which there are 100 possibilities), as well as **state-plus-digit codes** that activate on a specific digit when it follows a specific state (686 possibilities in our state machine). For example, code 2543 in MLP layer 2 activates on the 0 in the state 40 (e.g., 50-4**0**-59). This finding is notable because there are only 128 neurons in a given MLP layer, far lower than the total number of these features. Thus, the codebooks must disentangle features represented in a distributed manner across different neurons inside the network. (Anecdotally, the top-activating tokens for the neurons in these layers do not appear to follow any consistent pattern.)

We quantify this further with an experiment where we use state codes to *classify* states and compare them to the neuron with the highest precision at that state code's recall level. As shown in Figure 6a, codes have an average precision of 97.1%, far better than the average best neuron precision of 70.5%. These pieces of evidence indicate that codebooks can minimize the superposition problem in this setting. See Appendix C for additional details and experiments.

## 3.2 STEERING THE NETWORK BY ACTIVATING CODES

While these associations can provide hypotheses for code function, they do not provide causal evidence that codes causally influence the network's behavior. For this, interventional studies are necessary (Spirtes et al., 2000; Pearl & Mackenzie, 2018; Geiger et al., 2020; 2021). The state and state-plus-digit codes presented in Section 3.1 suggest a natural causal experiment: set the activated code in a given codebook to the code corresponding to another state and see whether the next token distribution shifts accordingly.[2] More specifically, let $\mathcal{C}^{(l)}(x_t)$ be the codebook at layer $l$ applied to

---

[2]This experiment is similar to what Geiger et al. (2020) call an interchange intervention, and more generally establish a *causal abstraction* over the neural network (Geiger et al., 2021).

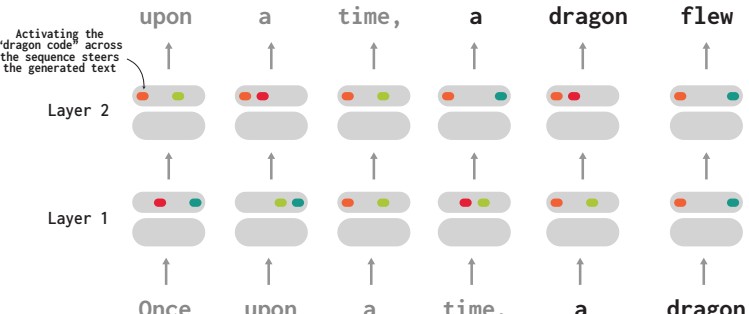

Figure 5: **Steering a language model with topic codes.** We identify several codes that activate on examples of a given topic (e.g., dragons). We then activate these codes at each generation step, producing generated text about that topic. See Table 10 for examples.

Table 2: **Codebook models are still capable language models.**. Asterisks (*) denote the base model we apply the codebooks to, while daggers (†) indicate the codebook models we analyze in the rest of the paper. We trained the other models to provide additional comparisons (see Appendix D.3 for more details, including on grouped codebooks.). All models have a codebook size of $C = 10k$. Note that the MLP 16-group $k = 8$ model is comparable to the attention $k = 8$ model because our model has 16 attention heads. While we use a pretrained TinyStories model as our base model, we also report metrics for a model we finetune to account for any subtle differences in data processing.

(a) TinyStories 1-Layer Model

| Language Model | Loss | Acc |
|---|---|---|
| *Pretrained | 1.82 | 56.22 |
| Finetuned | 1.57 | 59.27 |
| †Attn, $k = 8$ | 1.66 | 57.91 |
| MLP, $k = 100$ | 1.57 | 59.47 |
| MLP, grouped $16 \times (k = 8)$ | 1.60 | 59.36 |

(b) WikiText-103 410M 24-Layer Model

| Language Model | Loss | Acc |
|---|---|---|
| *Finetuned (Wiki) | 2.41 | 50.52 |
| Finetuned 160M (Wiki) | 2.72 | 46.75 |
| †Attn, $k = 8$ | 2.74 | 46.68 |
| Attn, $k = 64$ | 2.55 | 48.44 |
| MLP, $k = 100$ | 3.03 | 42.47 |
| MLP, grouped $16 \times (k = 8)$ | 2.73 | 46.16 |
| MLP, grouped $16 \times (k = 64)$ | 2.57 | 48.46 |

input token $x_t$. As we consider a $k = 1$ model, $C^{(l)}(x_t)$ returns a single code $c_t^{(l)} \in \mathbb{R}^d$. We replace this code with $\tilde{c}_t^{(l)}$, a code that activates when a different state is present. We then recompute the forward pass from that point and observe whether the network's next token distribution resembles the next token distribution for the new state.

In Figure 4a, we find that this is precisely the case—changing only the state codes in the MLP layers to a different state code shifts the next token distribution towards that other state, as measured by the Jensen-Shannon Divergence (JSD Lin, 1991), averaged over 500 random state transitions. This effect is even more substantial for the state-plus-digit codes, where changing the codes in the MLP layers makes the next-state distribution almost identical to that of the new state (Figure 4b). These results provide strong evidence that these codes perform the expected causal role in the network. Note that applying a similar perturbation to just a single MLP layer or all the attention layers causes a much smaller drop in JSD, indicating that this information is mainly stored across several MLP layers.

## 4 LANGUAGE MODELING

Next, we apply codebook features to language models (LMs) trained on naturalistic text corpora. We demonstrate the generality and scalability of our approach by training two models of different sizes on two different datasets. After describing the models we train and the training data, we follow the strategy described in Section 2.2 and identify hypotheses for the role of codes in the network. Then, we validate these hypotheses by steering the models through targeted activation of codes.

**Trained models** We finetune a small, 1-layer, 21 million parameter model on the TinyStories dataset of children's stories (Eldan & Li, 2023). We also finetune a larger, 24-layer 410M parameter

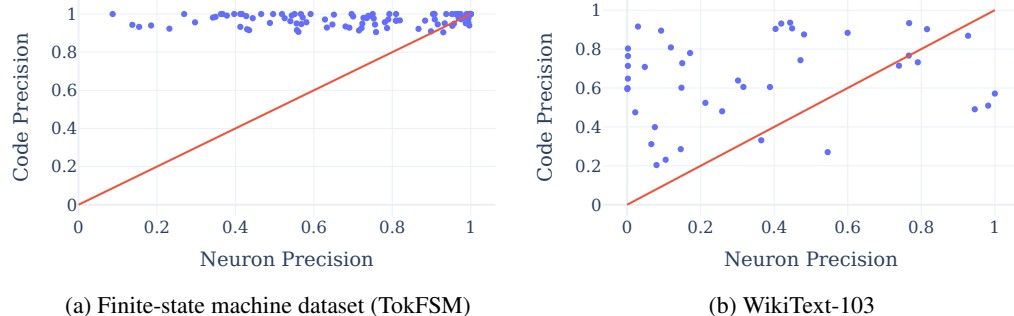

(a) Finite-state machine dataset (TokFSM)  (b) WikiText-103

Figure 6: **Codes are better classifiers of simple textual features than neurons.** *Y-axis*: precision of a given code at classifying a regular expression. *X-axis*: precision of the best neuron in the network, with a threshold chosen to match the recall of the code. *Red line*: $y = x$

model on the WikiText-103 dataset, consisting of high-quality English-language Wikipedia articles (Merity et al., 2016). See Appendix D for more training details.

**Codebook models are still strong language models**  Remarkably, despite the extreme bottleneck imposed by the codebook constraint, the codebook language models can still achieve strong language modeling performance. As shown in Table 2, codebook models can attain a loss and accuracy close to or better than the original models with the proper settings. In addition, the generations of the codebook look comparable to the base models, as shown in Table 10. Finally, in Appendix D.4, we profile the inference speed of these codebook models, showing how sparsity and fast maximum inner product search (MIPS) algorithms enable codebooks to run much more efficiently than the naive implementation of two large matrix multiplications.

**Generating hypotheses for the role of codes**  We also explore the interpretability of codes by looking at examples that the code activates on. In Table 11, we catalog codes that selectively activate on a wide range of linguistic phenomena, spanning orthography (e.g., names starting with "B"), word types (e.g., months of the year), events (e.g., instances of fighting), and overall topics (e.g., fire or football). Interestingly, codes for a particular linguistic phenomenon may not always activate on the words most relevant to that concept. For example, in our TinyStories model, we find a code that activates on mentions of fighting and violence might trigger on the word **the** but not the adjacent word **quarrel**. We suspect this may be because the network can store pieces of information in nearby tokens and retrieve them when needed via attention.

**Comparison to neuron-level interpretability**  As in Section 3.1, we would like to compare the interpretability of the codebook to neuron-level interpretability. While natural language features are more complex than the states in Section 3, we conduct a preliminary experiment comparing both neuron- and code-based classifiers to regular expression-based classifiers. We first collect a set of codes that appear to have simple, interpretable activation patterns (e.g., "fires on years beginning with 2"). We then created heuristic regular expressions targeting those features (e.g., `2\d\d\d`). Next, we compute the precision of the code classifier, using the regular expression as our source of truth. We then take the recall of our code classifier and search across all neurons, thresholding each at the same recall as the code and reporting the highest precision found. As Figure 6b demonstrates, codes are far better classifiers of these features than neurons on average, with over **30%** higher average precision. We defer additional details and discussion to Appendix D.7.

### 4.1 STEERING THE NETWORK BY ACTIVATING TOPIC CODES

As in Section 3.2, we would like to validate that codes do not merely fire in a *correlated* way with different linguistic features but that they have a *causal* role in the network's behavior. As an initial investigation of this goal, we study a subset of codes in the attention codebook model that appear to identify and control the *topic* discussed by a model. To identify potential *topic codes*, we use a

Table 3: **Activating topic codes causes the model to discuss those topics.** Percentage of generations that mention the topic before and after setting one or all codes in each attention head to the topic code. Numbers in (parentheses) indicate the number of activated topic codes. This number is smaller for the *all codes* condition because only one topic code will be activated if multiple topic codes are located in the same attention head.

(a) Wikitext

| Topic | Baseline Freq | Steered (one code) | Steered (all codes) |
|---|---|---|---|
| Video game | 2.5 | 55.0 $_{(18)}$ | **75.0** $_{(4)}$ |
| Football | 7.5 | 47.5 $_{(18)}$ | **95.0** $_{(8)}$ |
| Movie | 27.5 | 42.5 $_{(12)}$ | **90.0** $_{(5)}$ |
| Song | 20.0 | 32.5 $_{(17)}$ | **85.0** $_{(11)}$ |

(b) TinyStories

| Topic | Baseline Freq | Steered (one code) |
|---|---|---|
| Dragon | 2.5 | **65.0** $_{(8)}$ |
| Slide | 2.5 | **95.0** $_{(12)}$ |
| Friend | 42.5 | **75.0** $_{(9)}$ |
| Flower | 0.0 | **90.0** $_{(8)}$ |
| Fire | 2.5 | **100.0** $_{(16)}$ |
| Baby | 0.0 | **90.0** $_{(15)}$ |
| Princess | 40.0 | **87.5** $_{(14)}$ |

simple heuristic and select only codes that activate on more than $50\%$ of tokens in a given sequence.[3] Of these, we manually filter by looking at the activating tokens of these codes and choose only those that appear to activate frequently on other examples related to that topic.

To shift the output generations of the model, we then take an input prompt (e.g., the start-of-sequence token) and activate the topic codes in the model for every token of this prompt. Then, we sample from the model, activating the topic codes for each newly generated token. Unlike Section 3, our models here have $k > 1$. Thus, we explore two types of interventions: First, activating a **single** code in each codebook (replacing the code with the lowest similarity with the input) and second, replacing **all** activated codes in each codebook with $k$ copies of the topic code.[4] We use the attention-only codebook with $k = 8$ in our experiments. See Figure 5 for a graphical depiction.

Remarkably, activating the topic codes causes the model to introduce the target topic into the sampled tokens in a largely natural way. We show several examples of this phenomenon in Tables 4, 13 and 14. Interestingly, even though the topic code is activated at every token, the topic itself is often only introduced many words later in the sequence, when it would be contextually appropriate. We quantify the success of this method by generating many steered sequences and classifying the generated examples into different categories with a simple word-based classifier. The results, presented in Table 3, demonstrate that the steered generations mention the topic far more often, with almost all generations successfully mentioning the topic when all codes in a codebook are replaced. See Appendix D.8 for more details and additional generations. These interventions constitute meaningful evidence of how codebook features can enable interpretation and control of real language models.

## 5 RELATED WORK

**Mechanistic interpretability** Our work continues a long stream of work since the 1980s on understanding how neural networks operate, especially when individual neurons are uninterpretable (Servan-Schreiber et al., 1988; Elman, 1990) Recent work has continued these investigations in modern computer vision models (Olah et al., 2018; 2020; Bau et al., 2020b) and language models (Elhage et al., 2021; Geva et al., 2021), with special focus on the problem of understanding *superposition*, when many features are distributed across a smaller number of neurons (Elhage et al., 2022b). Recent work has investigated whether sparse dictionary learning techniques can recover these features (Yun et al., 2021; Sharkey et al., 2022), including the concurrent work of Bricken et al. (2023) and Cunningham et al. (2023). Our work shares similar goals as the above works. Codebook features attempt to make it easier to identify concepts and algorithms inside of networks by refactoring

---

[3]This heuristic is inspired by past work connecting activation patterns in frequency space to different linguistic phenomena (Tamkin et al., 2020)

[4]If $m > 1$ codes map to the steering topic in a given codebook, we replace the $m$ lowest-scoring codes in the first case and randomly select one code to replace all the codes in that codebook in the second case.

Table 4: **Example steered generations for TinyStories model.** More examples in Table 13

| Code Concept | # codes | Example steered generation |
|---|---|---|
| Dragon | 8 | **Once upon a time,** there was a little girl named Lily. She was very excited to go outside and explore. She flew over the trees and saw a big, scary dragon. The dragon was very scary. [...] |
| Flower | 8 | **Once upon a time,** there was a little girl Lily. She liked to pick flowers in the meadow. One day, she saw a big, green [...] |
| Fire | 16 | **Once upon a time,** there was a little boy named Timmy. Timmy loved his new toy. He always felt like a real fireman. [...] |
| Princess | 14 | **Once upon a time,** there was a little bird named Tweety. One day, the princess had a dream that she was invited to a big castle. She was very excited and said, "I want to be a princess and [...] |

their hidden states into a sparse and discrete form. We also show how codebooks can mitigate superposition by representing more features than there are neurons and that we can intervene on the codebooks to alter model behavior systematically.

**Discrete structure in neural networks** Our work also connects to multiple streams of research on incorporating discrete structure into neural networks (Andreas et al., 2016; Mao et al., 2019). Most relevant is VQ-VAE (van den Oord et al., 2017), which trains an autoencoder with a vector quantized hidden state (Gray, 1984). Our work also leverages vector quantization; however, unlike past work, we extend this method by using it as a sparse, discrete bottleneck that could inserted between the layers of any neural network (and apply it to autoregressive language models), enabling better understanding and control of the network's intermediate computation.

**Inference-time steering of model internals** Finally, our work connects to recent research on steering models based on inference-time perturbations. For example, Merullo et al. (2023) and Turner et al. (2023) steer networks by adding vectors of different magnitudes to different layers in the network. Our work supports these aims by making it easier to localize behaviors inside the network (guided by activating tokens) and making it easier to perform the intervention by substituting codes (so the user does not have to try many different magnitudes of a given steering vector at each layer).

We include an extended discussion of related work, including the relative advantages of codebooks and dictionary learning methods in Appendix E.

## 6 DISCUSSION AND FUTURE WORK

We present *codebook features*, a method for training models with sparse and discrete hidden states. Codebook features enable unsupervised discovery of algorithmic and linguistic features inside language models, making progress on the superposition problem (Elhage et al., 2022b). We have shown how the sparse, discrete nature of codebook features reduces the complexity of a neural network's hidden state, making it easier to search for features and control a model's behavior with them.

Our work has limitations. First, we only study Transformer neural networks on one algorithmic dataset and two natural language datasets; we do not study transformers applied to visual data or other architectures, such as convolutional neural networks, leaving this for future work. In addition, we only explore topic manipulation in language models; future work can explore the manipulation of other linguistic features in text, including sentiment, style, and logical flow.

Ultimately, our results suggest that codebooks are an appealing unit of analysis for neural networks and a promising foundation for the interpretability and control of more complex phenomena in models. Looking forward, the sparse, discrete nature of codebook features should aid in discovering circuits across layers, more sophisticated control of model behaviors, and making automated, larger-scale interpretability methods more tractable.[5]

---

[5] See Appendix F for an extended discussion of applications and future work.

## REPRODUCIBILITY STATEMENT

We release our codebase and trained models to enable others to easily build on our work. Additionally, Sections 2 to 4 and appendices A, C and D describe the specific experimental details and settings we used to carry out our experiments.

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

## A  GENERAL TRAINING AND OPTIMIZATION DETAILS

Here, we provide some additional training details relevant to all experiments.

**Layer norm**  We apply layer norm to the input activations of the codebooks, which we found improved accuracy and stability.

**Optimizer hyperparameters**  Unless otherwise specified, we use the Adam optimizer (Kingma & Ba, 2014) with learning rate 5e-4 and default values of $\beta_1 = 0.9, \beta_2 = 0.99$. For experiments using learning rate decay this refers to the peak learning rate; we spend 5% of training on a linear warmup to the max learning rate and the rest on a linear decay to 0. We did not find a benefit to using weight decay in our experiments. We also found no benefit to using k-means initialization of the codebooks.

**Training hyperparameters**  We train for 15k steps for most experiments. For the TinyStories datasets, we train for 100k steps. The sequence length for WikiText-103 is 1024, and for TinyStories it is 512. Depending on the model, we use a batch size of 64 to 256 and between 1-4 A100 GPUs. By default, codebooks have $C = 10k$ codebook size unless otherwise specified.

## B  CODEBOOKS AS INFORMATION BOTTLENECKS

Codebooks are information bottlenecks: they limit the bits of information that can be transmitted from a given layer into the rest of the network. Intuitively, they force the network to represent its activations as a choice of $k$ distinct, unordered codes out of a vocabulary size of $C$. This fact enables us to compute the *channel capacity*, or number of bits the codebook can transmit each forward pass: $\lceil \log_2 \binom{C}{k} \rceil$. In Table 5, we present the channel capacity of various codebooks of size 10,000 with values of $k \in [1, 8, 100]$. We also compare this with the channel capacity of a standard 16-bit activation with size 1024 hidden state, as well as quantized 4-bit vectors. We observe that even the $k = 100$ case transmits far fewer bits than even a 4-bit quantized 1024-dimensional vector.

Table 5: Comparison of information content for different information bottlenecks.

| Scenario | Bits Transmitted |
|---|---|
| 1024-dimensional 16-bit vector | 16384 |
| 1024-dimensional 4-bit vector | 4096 |
| 1 code from codebook of size 10,000 | 14 |
| 8 codes from codebook of size 10,000 | 91 |
| 100 codes from codebook of size 10,000 | 804 |

## C  FINITE STATE MACHINE EXPERIMENTS

This section presents additional details and experiments for the finite state machine (FSM) domain.

### C.1  TOKFSM TRAINING HYPERPARAMETERS

We use a constant learning rate of $1e - 3$ with a batch size of 512 and train the models for $20,000$ training steps. Note that the architecture used in Section 3 uses parallel attention and MLP blocks, following (Black et al., 2022).

### C.2  DEAD CODES

After training the models, we notice that many codes in the model do not activate at all on the eval set; we refer to these as *dead codes*, and the opposite as *active codes* (Yu et al., 2021). We report the number of active codes for each component of the $k = 1$ Attn+MLP codebook model in Table 7,

computed over an evaluation set of 10240 samples of sequence length 128. While many codes end up dead, we find that starting training with fewer codes leads to worse accuracy than training with more codes than needed, suggesting some role for dead codes in the codebook optimization process.

## C.3 ADDITIONAL OBSERVATIONS FROM ACTIVATING TOKENS

Although the strongest form of evidence we consider are the causal intervention experiments in Section 4.1, we briefly overview a range of different types of codes we identify through qualitative observation:

- Codes in MLP layer 0 (the first MLP layer), which activate on each different token
- Codes in MLP layers 1, 2, and 3, which activate on bigrams corresponding to different states of the FSM (e.g., 42, 59, 29), only on the second digit of a state (*state codes*)
- Codes in MLP layers 1, 2, and 3, which activate on trigrams: (e.g., 823, 182), only on the first digit of a state (*state-plus-digit codes*)
- In many cases, several different states (or state-plus-digits) activate the same code. In Appendix C.4, we show that these state groups have much more similar next-token distributions than average codes and provide potential interpretations for this phenomenon.
- Codes that activate on bigrams or trigrams, regardless of which digit they are present on
- Codes in several attention heads, which activate on states *beginning* with a specific digit (e.g., $51, 52, 53\ldots$)
- Codes that do not appear to fire on any discernible pattern.

From these points of anecdotal evidence, we make several broader observations:

1. The network learns codes that fire in association with useful high-level features of the input space, e.g., when a given FSM state is present

2. Individual features are not necessarily isolated to a single point in the network; multiple places may represent the same piece of information, as (Bau et al., 2020b) found in a computer vision context.[6]

3. It is possible for the behavior of a given layer to be *position dependent*—that is, the network can store different information in the same layer depending on the position in the sequence. For example, the same MLP layer may hold different information when the input token is the first digit vs. when it is the second digit of a state. Thus, absolute statements that certain layers or attention heads "store concept X" warrant caution, as this layer's function could be contextually dependent.

4. Sometimes, the network forms representations that seem to admit a meaningful interpretation but do not immediately appear useful to the network. For example, it initially seems useless to have a code that activates based on states that share the same first digit (e.g., 51, 52, 53, ...) as these states are unrelated. It may be possible this code is used as part of a *circuit* to identify an FSM state in a future layer, or perhaps it is simply a vestigial or spandrel feature (Gould & Lewontin, 1979; Gould, 1997).

## C.4 ANALYSIS OF CODE PURITY IN THE FINITE-STATE-MACHINE MODELS

The TokFSM dataset from Section 3 was designed such that we know the exact number of features in the data, permitting us to understand how the representation of these features changes across the network. In Figure 8, we plot the fraction of codes that are *pure* at each layer, meaning they activate only on a single state (in the case of *state codes*) or state and first digit (in the case of *state-plus-digit* codes). We compute these statistics over all valid combinations of two- or three-digit starting sequences. We see very high levels of purity for both sets of codes. The high purity of the codes at the first layer demonstrates that codebook training has mostly resolved the superposition problem at the first layer.

---

[6]We suspect it may be possible to detect these families of codes by computing co-occurrence statistics, but we leave this to future work.

Table 6: Example Code Activations for the **TokFSM** dataset. The **bolded** digits indicate the token positions that activated the given code. Hyphens (-) are added between each state for readability but are not presented to the model. MLP codes are written in the form `layer.code-id`, while attention codes are written in the form `layer.head.code-id`. More activations are available at [redacted for anonymity] .

| Code | Interpretation | Example Activations |
|---|---|---|
| MLP 0.2523 | **1** digit | 3**1**-83-40-87-80-78-38-76-03-86-**1**7-97-76-09-**1**5 |
| | | **1**0-57-62-43-92-3**1**-83-82-23-65-94-33-23-49-4**1** |
| | | **1**9-83-3**1**-73-29-47-04-**1**5-77-05-79-23-47-89-95 |
| MLP 1.2527 | 48**9** trigram (either pos.) | 86-04-8**9**-80-17-03-40-74-24-09-93-35-59-61-49 |
| | | 40-46-50-38-47-04-8**9**-80-91-82-94-33-41-77-59 |
| | | 18-94-55-55-48-24-68-48-**9**0-43-97-50-74-77-59 |
| MLP 2.2543 | 4**0** bigram (2nd pos.) | 80-04-70-50-4**0**-59-07-73-28-02-71-54-31-62-40 |
| | | 74-05-13-72-95-66-52-31-98-20-88-4**0**-59-22-19 |
| | | 4**0**-46-44-01-88-66-51-14-41-57-18-84-89-60-51 |
| Attn 1.2.3207 | Tokens after 44 bigram | 44-**2**7-74-05-59-64-67-72-42-93-35-09-67-39-96 |
| | | 44-**2**7-74-05-22-65-98-75-83-20-00-60-80-57-94 |
| | | **7**7-69-28-02-34-46-52-72-94-18-84-12-16-64-4**6** |
| Attn 2.0.3044 | Tokens on or after 59 | 74-05-5**9**-64-67-72-42-93-35-09-67-39-96-07-96 |
| | | 88-40-5**9**-**2**2-19-33-31-93-42-53-75-94-33-31-76 |
| | | 87-14-40-59-**2**4-72-86-04-30-04-81-56-01-17-30 |

The code purity declines in higher layers as the model forms its prediction of the next token. Why is this? As Figure 9 demonstrates, when two different states activate the same code, they tend to have much more similar next-token distributions. Specifically, the next-token distributions of trigram states that activate the same code (red bars) are much smaller than those of random pairs of trigram states (blue bars). This result suggests that states are merged when they share a similar next-token distribution. We speculate that codes merge later in the network as the network shifts from identifying the state to forming its prediction of the next token, as previous work has also speculated (Elhage et al., 2022a).

In general, we believe that better understanding when two concepts share a code is a fruitful avenue for future study.

Table 7: **Number of active codes in** $k = 1$ **attention + MLP codebook model trained on Tok-FSM**. Each codebook has 10,000 codes; most of the codes in each codebook are not active by the end of training.

| Layer | Head 0 | Head 1 | Head 2 | Head 3 | MLP |
|---|---|---|---|---|---|
| 0 | 40 | 45 | 41 | 49 | 11 |
| 1 | 293 | 367 | 657 | 460 | 1027 |
| 2 | 1482 | 3071 | 1103 | 1499 | 943 |
| 3 | 690 | 282 | 315 | 1233 | 247 |

## C.5 ABLATION EXPERIMENTS

We perform several ablation studies to identify the importance of different elements of our training method. Specifically, we compare the next-token accuracies of several families of models, including the TinyStories one-layer model, the 4-layer TokFSM model, and the 24-layer wikitext model. For each model, we present the accuracies for 1) the attention codebook model presented in the paper, 2) the same model but with a random initialization as opposed to the pretrained model, and 3) a codebook model where the model parameters were frozen and only the codebook parameters were trained, and 4) a model where only the codebook parameters were trained, and they were trained

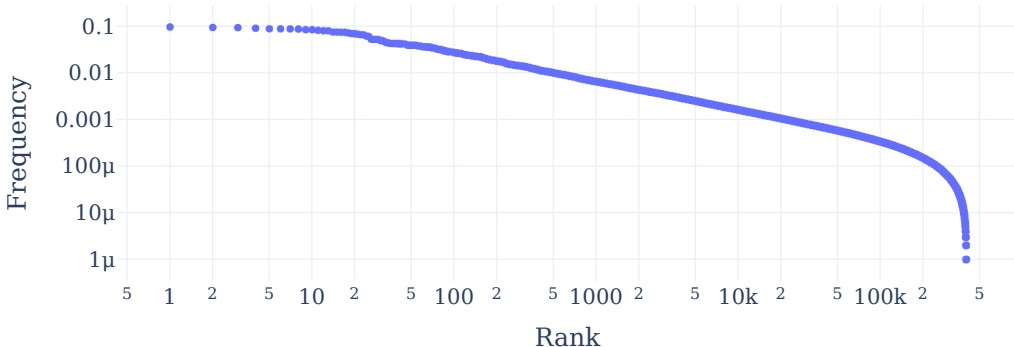

Figure 7: **Code activation frequencies appear to follow a power law** Frequency of code activations by rank from TinyStories 1-layer attention-only codebook model. The x-axis denotes the rank of the code in terms of frequency on a subset of the training set. We observe that most codes activate very rarely, while a long tail of codes activate very frequently.

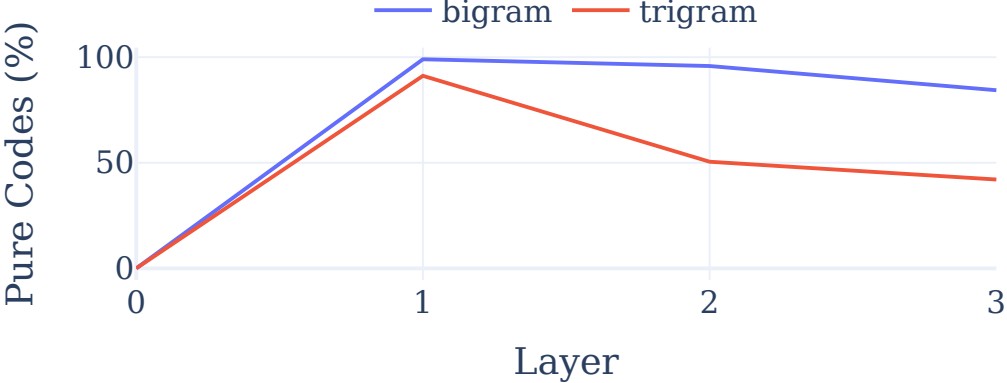

Figure 8: **Codebook training overcomes the superposition challenge in the first layer.** We plot the fraction of codes which are *pure* at each layer, meaning they activate only on a single state (in the case of bigrams) or state + first digit (in the case of trigrams). We see very high levels of purity for both bigram and trigram models. Because the number of hidden states is 128, and there are 1000 trigram combinations for the model to learn, the network cannot allocate each state to a different neuron. The high purity of the codes demonstrates that codebook training has mostly resolved the superposition problem at the first layer. Code purity declines in higher layers as the model forms its prediction of the next token (see Figure 9). Experiment performed on the MLP codebooks of the $k = 1$ Attn + MLP codebook TokFSM model over all 100 and 1000 possible combinations of the first two and three digits, respectively.

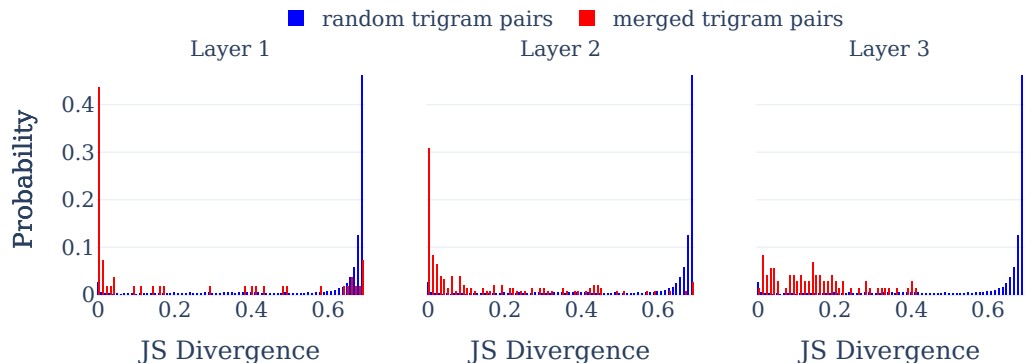

Figure 9: **When two different states activate the same code, they tend to have much more similar next-token distributions.** We find that the next-token distributions of trigram states that activate the same code (red bars) are much smaller than those of random pairs of trigram states (blue bars). This result suggests that states are merged when they share a similar next-token distribution. *X-axis*: Jenson-Shannon Divergence (JSD) between next-token distributions of different states. The JSD is a measure of the distance between probability distributions).

with only the autoencoding portion of the loss. The results of these experiments are presented in Table 8. Broadly, we find that all components are necessary for strong performance, although we do not exhaustively tune hyperparameters for each ablation.

Table 8: **Ablation studies.** Next-token accuracy (for TinyStories and WikiText-103) and next-state transition accuracies (for TokFSM) across various ablation studies. *Legend*: **Attn CB**: Codebook applied to the attention layers. **Random Init**: Codebooks applied to a randomly-initialized model instead of a pretrained model (then finetuned end-to-end as usual). **Train Only CB**: Train only the codebook layers with the original loss while keeping the base model frozen. **Only AE Loss**: Only apply the autoencoding loss to the codebooks; do not update the model parameters. **Attn + MLP CB** Codebooks applied to the attention and MLP codebooks simultaneously.

| Model | Attn CB | Random Init | Train Only CB | Only AE Loss |
|---|---|---|---|---|
| TinyStories-1L | 57.91 | 55.67 | 47.08 | 51.73 |
| FSM-4L | 96.39 | 52.35 | 58.48 | 43.44 |
| WikiText-103-24L | 46.16 | 38.53 | 31.22 | 28.35 |

# D  LANGUAGE MODEL EXPERIMENTS

## D.1  1-LAYER TINYSTORIES MODEL

We train a small, 1-layer 21 million parameter transformer on the TinyStories dataset of children's stories, constructed by prompting a language model (Eldan & Li, 2023). We train for 100k steps with a batch size of 96, with learning rate warmup of 5% and linear cooldown to 0. We start by loading the 21M pretrained model from the TinyStories paper (Eldan & Li, 2023). We train two models: one with the codebook affixed to each of the heads of all the attention layers and one to both the attention heads and MLP layers (Figure 2).

In Figure 7, we plot the distribution of code activation frequencies for the 1-layer TinyStories $k = 1$ Attn + MLP model. We find a very unequal distribution of use of the codebooks, with a small number of codes activated extremely frequently and many others activated hardly at all. This distribution is reminiscent of the Zipfian distribution known to characterize phenomena such as word frequency in natural language (Kingsley Zipf, 1932).

## D.2   24-LAYER WIKITEXT-103 MODEL

We also train a larger, 24-layer 410M parameter model on the WikiText-103 dataset, consisting of high-quality English-language Wikipedia articles. We finetune for $20,000$ steps with a batch size of 24 and learning rate warmup and cooldown. For a pretrained model, we use the Pythia 410m parameter model, trained on the Pile dataset with deduplication (Biderman et al., 2023). The model has 16 attention heads, with a hidden size of 1024. We again train two variants of codebook models here, with codebooks on every attention head and codebooks on every MLP block.

## D.3   COMPARING THE PERFORMANCE OF CODEBOOK AND BASE MODELS

Here, we provide more details on the models trained in Table 2. Most model names in the table are self-explanatory; for example, `MLP, k=100` indicates a model with codebooks on the MLP layers with a $k$ of 100. The only exceptions are as follows:

**Finetuned 160M (Wiki)**   The largest base language model we finetune is a 410M parameter 24-layer model from the Pythia series of models (Biderman et al., 2023), finetuned on the WikiText-103 dataset (Merity et al., 2016). To explore how much codebooks reduce the performance of language models, we also finetune the next smallest model in the series: a 160M parameter 16-layer model. As we see, the language modeling accuracy of the Attn $k = 8$ model is comparable to this smaller model, and the Attn $k = 64$ model falls squarely in between the 160M and 410M parameter models.

**MLP, grouped $16 \times (\mathbf{k = 8}$ or $\mathbf{64})$**   The MLP codebook layers broadly seem to attain lower performance than the attention layers. Moreover, we found diminishing returns to increasing the value of $k$ for this layer. We observe that we can attain higher performance for these layers by splitting the MLP layer activations into several equal-sized chunks (16 in our case) and training a smaller codebook independently on each chunk, as in product quantization (Jegou et al., 2010). We refer to this method as "grouped codebooks."

All models except the grouped MLP codebook model are trained with the same hyperparameters. We found that the grouped MLP codebook model achieved 4-5% higher accuracy and trained more stably if we used a 10x higher learning rate on the codebook parameters than the default learning rate (which was used for the language model parameters). We suspect the combination of grouped codebooks and higher learning rates on the codebook parameters may be helpful when applying codebooks to higher-dimensional layers. While we suspect the primary benefit of grouped codebooks is in aiding optimization, an interesting direction for future work is whether they improve expressivity or interpretability of the resulting codebooks.

## D.4   CODEBOOK MODELS STILL HAVE USABLE INFERENCE SPEED

The codebook modules at each attention head add parameters and computation to the model. While this results in higher latency, the resulting model is still usable for real-time inference. Moreover, inference can be sped up an additional amount through fast maximum inner product search (MIPS) algorithms such as FAISS, which are faster than computing the matrix multiplication explicitly (Johnson et al., 2019). In Table 9, we show that the codebook models show a significant decrease in the number of generated tokens per second (between 34% and 69% slowdown). However, this decrease is significantly lower when FAISS is used. A decrease in latency may be acceptable in exchange for increased interpretability or control, and we expect further optimizations (e.g., approximate MIPS algorithms, custom kernels) to continue to close this gap.

## D.5   EXAMPLE LANGUAGE MODEL GENERATIONS

We display example generations from both language models in Table 10.

## D.6   ACTIVATING TOKENS

We present examples of activating tokens for both language models in Table 11

Table 9: **Maximum inner product search algorithms can close much of performance gap between codebook and tranditional models.** Performance Comparison of Models with Different Parameters. Computed on an A100 40GB GPU, with a batch size of 64 and over 100 batches.

(a) 70m Parameters

| Model | Tok/s | $\Delta$ FAISS | $\Delta$ Base |
|---|---|---|---|
| Base | 57.5 | | |
| CB w/ FAISS | 37.4 | 34.2% | -34.9% |
| CB no FAISS | 27.9 | | -51.5% |

(b) 410m Parameters

| Model | Tok/s | $\Delta$ FAISS | $\Delta$ Base |
|---|---|---|---|
| Base | 14.8 | | |
| CB w/ FAISS | 7.2 | 56.2% | -51.5% |
| CB no FAISS | 4.6 | | -68.9% |

Table 10: Example generations from language models. The prompts are highlighted in bold. While the factuality of the completions is unreliable for all models, all models generate largely grammatical text.

| Language Model | TinyStories 1-Layer Model | WikiText-103 Model |
|---|---|---|
| Base | **Once upon a time** there was a little boy named Timmy. Timmy loved to play outside in the rain. He would jump in puddles and splash around. One day, Timmy saw a big puddle in the park. He jumped in it and got all wet.[...] | **The war was fought** against the Ottoman Empire and the Kingdom of Hungary. The Ottoman Turks, their king, and several of their princes were killed and many more captured, and the kingdom was divided among the Hungarian monarchs ; [...] |
| Codebooks (Attn) | **Once upon a time**, there was a little girl named Lily. She loved to play with her toys and her friends. One day, Lily's mom told her that they were going to buy a new toy. Lily was very excited and asked, "Can I play with your toys, please?"[...] | **The war was fought** by France and the British Empire, and by the Axis powers. With the exception of the Italians and Americans, whose armies won the war against the Axis Powers, the victorious Allies suffered the most of the war, a terrible defeat on both fronts. [...] |
| Codebooks (MLP) | **Once upon a time**, there was a little boy named Timmy. Timmy loved to play with his toy cars and trucks. One day, Timmy's mom took him to the store to buy a new toy. Timmy saw a big red truck and asked his mommy if they could get it, but she said they had to wait until they got to the store. | **The war was fought** between the United States and France. The French responded by launching an invasion of the Allied continent in June 1917 with the aim of defeating the Allied armies in northern France. [...] |

Table 11: Example Code Activations for the **TinyStories** and the **WikiText-103** dataset. The **bolded** word indicates the token positions that activated the given code. **Note that the concept may be near but not directly at the activated token.** MLP codes are written in the form `layer.code-id`, while attention codes are written in the form `layer.head.code-id`. At symbol (@) delimiters present in WikiText-103 data have been omitted for readability. More activations are available at [redacted for anonymity] .

(a) WikiText-103

| Code | Interpretation | Example Activations |
|------|----------------|---------------------|
| 7.12.7884 | Months (after preposition) | at Toulon in **August** The ship began trials [. . . ] and spent three weeks in **September** attached to
14 : 30 on 7 **December**. The division had the [. . . ] a major attack until 8 **December**
on **August** 31, a Utah [. . . ] On **September** 1, 1987 |
| 4.15.6101 | Evaluative words | Initially , the New Zealand attack progressed **well**
Superman from the main timeline is **successfully** teleported into
only HWMs evaluated as "**excellent**" are used by NHC |
| 1.9.295 | Names starting with 'B' | In one account from the Bah**amas** , a mating pair ascended

while John and Roy B**oulting** noted that [...]
B**ocks**car, sometimes called B**ock**'s Car, is the name of the United States Army Air Forces B-**29** bomber |
| 4.14.4742 | Years in 2000s | As of **2011** , the International Shark Attack File lists
In **2014** , a study at the University of Amsterdam with
Fabian Cancellara kicked off his **2010** campaign with an overall victory at the Tour of |
| 9.3.3727 | Square Units | Atlanta encompasses 134.0 **square miles** (347.**1**km**2**)
it covered more than 55 square metres (590 **sq** ft)
6 percent or 101,593 **square** kilometres (39,**225 sq** mi) of [...] |

(b) TinyStories

| Code | Interpretation | Example Activations |
|------|----------------|---------------------|
| 0.2 | Fighting | The two cats started to **quarrel loudly** over the bone
They ran around the house, fighting over **the** thread
But then, they got into a fight **over** who got to play with the toy |
| 0.3 | Negative emotions | He feels angry and **scared**. He tries to catch the boat, but it
She started to feel **nervous** because she thought she wouldn't be able to
Lily and Tom felt **fearful**. They did not like storms. |
| 0.6 | "You" dialogue | The dragon smiled and said, "**You are** too small. It's not possible."
The happy fish thanked her and said "**You must** be very persistent to complete this task.
John smiled and said, "**You won**! You were really fast." |
| 1.2 | Fire | The fire **spread to** the **cans and bottles and** made more explosions. The garage was full of smoke
Lily knew that fire could be dangerous and she **always** remembered to be careful **when** playing with matches or **li**ghters.
Mom hugged them and said, "I know, but **fire** is **not a** toy. **It can hurt you and** the plants **and** animals. |
| 5.3 | Discovered/found | Lily found a delicate flower in the garden and **showed** it to her sister.
had discovered an amazing reef and helped a turtle in **need**.
One day, Tom and Mia found a ball in **the** hut. |

Table 12: Regular expressions used to measure topic steering for the text generated by the models.

(a) Wikitext

| Topic | Regex |
|---|---|
| Football | football\| soccer\| goal\| stadium\| fifa\| player\| trophy\| league |
| Movie | movie\| tv\| television\| film\| media |
| Video Game | game |
| Song | song\| music\| mtv |

(b) TinyStories

| Topic | Regex |
|---|---|
| Dragon | dragon |
| Slide | slide |
| Friend | friend |
| Tom & Sam | tom\| sam |
| Flower | flower |
| Fire | fire |
| Baby | baby |
| Princess | prince\| crown\| king\| castle |

### D.7 ADDITIONAL NOTES ON NEURON-LEVEL INTERPRETABILITY EXPERIMENTS

We briefly note two caveats to this preliminary experiment. First, regular expressions are not perfect proxies for the features we care about (e.g., our regular expression for countries only includes some countries or ways of spelling each country). Thus, these precision scores likely underestimate each classifier's true precision. Second, we note a potential bias in the experimental protocol due to developing the regular expressions for codes that admit a meaningful interpretation. This could result in a slight bias in favor of the code classifiers. However, we also exhaustively search over all 410 million neurons in the network to find the best performer, which mitigates this bias. The complete list of regexes we use is available in our codebase.

### D.8 LANGUAGE MODEL STEERING EXPERIMENTS

We present additional language model steering results in Table 13.

Note that while we use the MLP codes to steer the TokFSM model, we use the attention codes to steer the WikiText model. The reason for using different codes here is because we are trying to control different aspects of the sequence/text in each model. In the TokFSM environment, we are trying to alter the prediction of an individual state or token. We find codes in the MLP layers are most associated with these single tokens. For the language modeling experiments, we are trying to alter the global topic of a generation. Topics typically manifest across many tokens, rather than a single token, and we find the attention layers are most associated with these features. However, we believe it is quite possible that for more local linguistic features (such as word choice) editing the MLP codes in a language model may prove to be the best way to edit the model's behavior.

### D.9 HOW DOES VARYING THE NUMBER OF CODES INFLUENCE HOW WELL THE MODEL CAN BE STEERED?

To steer the language model, we activate a number of topic codes discovered throughout the network. In Figure 10 we plot how changing the number of codes activated increases the rate at which the topic is introduced to the generation. We observe a general increase in the steering rate as the number of topic codes activated increases.

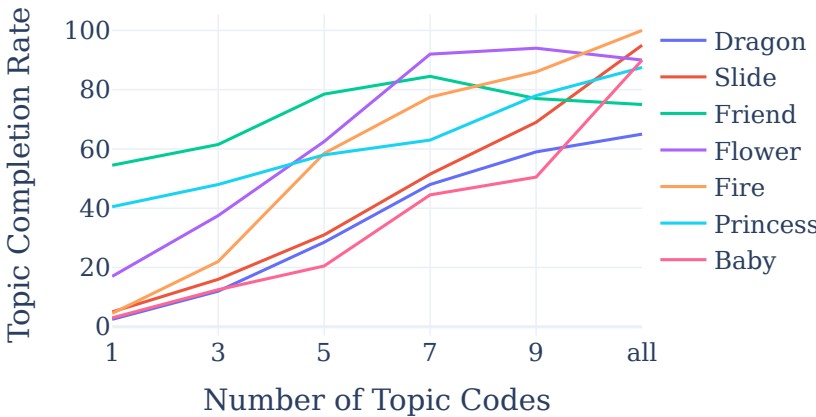

Figure 10: Increasing the number of topic codes activated increases the rate at which the language model is successfully steered to the desired topic.

Table 13: Steering the TinyStories language model with topic codes. While the language model generations are often incoherent, the topics are largely introduced in grammatical ways

| Topic | Codes | Steered generations |
| --- | --- | --- |
| Null (default generation) | - | **Once upon a time,** there was a little girl named Lily. She loved to play with her toys and eat yummy food. One day, she went to the store with her mommy to buy some apples. She saw a big basket and wanted to buy it. Her mommy said, "Let's count the apples and buy the yummiest apples!" Lily was so happy and said, "Yes, please!" |
| Dragon | 8 | **Once upon a time,** there was a little girl named Lily. She was very excited to go outside and explore. She flew over the trees and saw a big, scary dragon. The dragon was very scary. But Lily knew that things were not real and she would hurt her. |
| Slide | 12 | **Once upon a time,** there was a big cake. He wanted to escape and explore the world, but he was scared. He thought and thought about how to use the slide. He asked his mom for some help. |
| Friend | 9 | **Once upon a time,** there was a little girl named Lily. She had a friend named Sue. Sue was a nosy cat. She liked to play with her brother Tom. One day, they went to the park and saw a big dog. The dog was not nice. |
| Flower | 8 | **Once upon a time,** there was a little girl named Lily. She liked to pick flowers in the meadow. One day, she saw a big, green caterpillar on a leaf. She wanted to take it home and sell it to someone else. As she picked the flower, it started to bloom and made it look pretty. |
| Fire | 16 | **Once upon a time,** there was a little boy named Timmy. Timmy loved his new toy. He always felt like a real fireman. One day, Timmy's mom made him some hot soup and gave him some medicine to help his mommy feel better. Timmy was scared that the fire would be gone, but he didn't feel happy. |
| Baby | 15 | **Once upon a time,** there was a little girl named Lily. She loved going to the gym with her mommy. One day, Lily's mom asked her to help put the baby in the crib. |
| Princess | 14 | **Once upon a time,** there was a little bird named Tweety. One day, the princess had a dream that she was invited to a big castle. She was very excited and said, "I want to be a princess and ride the big, pretty castle!" |

Table 14: **Steering language model generations with topic codes**. Full table available at [redacted for anonymity] . At symbol (@) delimiters present in WikiText-103 data have been omitted for readability.

| Topic | Codes | Original generations | Steered generations |
|---|---|---|---|
| Video game | 18 | **The war was fought** on two fronts. The war was initiated in 1914 between Austria-Hungary and Serbia, when the Entente Powers signed a treaty of friendship between the two countries. In October 1914, Tschichky was sent to defend the German Empire' | **The war was fought** on both sides, and was only the second game to deal with one-on-one battles, following SimCity 2D Blade II. The game was released to critical acclaim, with praise particularly directed to the new console |
| Football | 18 | **The war was fought** on two fronts. The war was initiated in 1914 between Austria-Hungary and Serbia, when the Entente Powers signed a treaty of friendship between the two countries. In October 1914, Tschichky was sent to defend the German Empire' | **The war was fought** in its first forty years. In the summer of 1946, the Cardinals of the All-America Football Conference (AAFC) were rapidly becoming the favorites for NFL Hall-of-Fame coach Jim Mora, who had |
| Movie | 12 | **The novel was published in** November 2009 by MacChinnacle, a London publishing house. The book's publishers, Syco, published the book in the United Kingdom and the United States on 1 November 2009. The book received generally positive reviews from critics, who praised the | **The novel was published in** the United States and Canada. The film was directed by Joe Hahn and stars Steven Spielberg as Lucas, Neil Patrick Harris, and Jude Lawder as Lucas's best friend, Jonathan Miller. The plot follows a character (Lucas |
| Song | 17 | **The team won** their first ever Grand Prix and the first since the 1990 season. The team finished in third place behind Williams and Ralf Schumacher, with the Ferraris of David Coulthard and Jarno Trulli finishing in the top three. | **The team won** the Grammy Awards for Best Gospel Album. = = Background = = In 2004, The Dream released their third studio album, The Beacon Street Collection, which produced the singles "HOV Lane" and "Wishing Machine |

# E   EXTENDED DISCUSSION OF RELATED WORK

In this section, we review related work and attempt to describe in more detail the design decisions behind codebook features and how these lead to different tradeoffs compared to other approaches. We focus on several subareas most relevant to our current work, with a particular focus on dictionary learning methods, leaving more general overviews of interpretability research to prior surveys (Rogers et al., 2021; Bommasani et al., 2021; Madsen et al., 2022).

## E.1   SPARSE CODING AND SPARSE DICTIONARY LEARNING

Sparse coding, also known as sparse dictionary learning, is a well-studied research area with applications in machine learning, neuroscience, and compressed sensing (Kanerva, 1988; Olshausen & Field, 1997; Lee et al., 2006; Candes et al., 2006; Donoho, 2006; Rozell et al., 2008). The typical objective in sparse coding is to learn a fixed set of vectors, known as *atoms* or *dictionary elements*; given this set of vectors, one should be able to represent a given input as a sparse linear combination of these vectors. Sparse coding methods have been applied to various problems in machine learning, including in computer vision (Elad & Aharon, 2006) and natural language domains (Zhu & Xing, 2012; Arora et al., 2018).

Dictionary learning methods have recently seen renewed interest as an interpretability approach for neural networks (Yun et al., 2021; Wong et al., 2021). One reason for this is the *superposition problem*: to represent more feature directions than neurons, some neurons will be activated for multiple different features (Yun et al., 2021; Elhage et al., 2022b). For example, one family of approaches trains a wide autoencoder with a sparsity penalty. The width of the autoencoder is made greater than the size of the input activations (producing an *overcomplete basis*); by regularizing the activations of the autoencoder to be sparse, the dimensions of the autoencoder appear to correspond to more disentangled features (Yun et al., 2021; Sharkey et al., 2022; Bricken et al., 2023; Cunningham et al., 2023).

Codebook features share important similarities with dictionary learning approaches: for example, both approaches learn a codebook of elements larger than the number of input neurons and attempt to activate a small fraction of that basis on each forward pass. However, a significant conceptual difference between codebook features and dictionary learning is their implicit choice of *how features are represented* inside of neural networks:

### E.1.1   FEATURES-AS-DIRECTIONS

Recent dictionary learning approaches typically start from an assumption we might call *features-as-directions*: features the network learns are represented as continuous vectors along a *direction* in activation space. This assumption is substantiated by prior work on interpretability (Kim et al., 2018; Olah et al., 2018), and has the benefit that the magnitude of the vector along that direction corresponds to the strength of the feature or the probability of the feature existing in the data. However, the *feature as directions* assumption also faces some challenges:

**A direction can hold multiple features**   First, a single direction can theoretically represent multiple distinct features. For example, the positive and negative magnitudes of a direction could each hold a different (mutually exclusive) feature, which could be extracted by outgoing weights of $1$ and $-1$, respectively, in combination with a ReLU activation. More complex encodings of multiple features within a single direction are possible with bias terms and activation functions. For example, a network could detect whether a feature along direction $x$ has low, medium, or high magnitude by computing softmax$(x, 2x - 1, 5x - 7)$; the first dimension is greatest when $x < 1$, the second when $1 < x < 2$ and the third when $x > 2$.

**Continuous features can be challenging to interpret**   Second, the continuous and graded nature of feature directions can make them challenging to interpret: does an increase in the magnitude of one feature mean the network is more confident the feature is present, or merely that the strength of the feature is stronger in the input? If an input activates a feature at magnitude 0.52, or more strongly than in 90% of inputs, does this mean the feature is present? The same factors also make

it challenging to compare the strengths of different features without understanding how the network weights process each of them.

**Smuggling of information**  Another difference between codebook features and dictionary learning approaches is the contrast between soft and hard sparsity. Recent dictionary learning approaches train an L1-regularized autoencoder (Sharkey et al., 2022). This method causes the hidden activations of the autoencoder to have a small number of entries with a high magnitude but does not force the model to set the other features to be exactly zero. Past work has suggested that important information can be "smuggled" via low-magnitude activations (Elhage et al., 2022a), making it challenging to be confident that the interpretable features found by a dictionary learning approach are fully capturing the information a network is detecting in the input.

### E.1.2 FEATURES-AS-POINTS

In contrast, codebook features embody a view of *features-as-points*. For example, an activated code is simply a vector of fixed magnitude that is added to the output of the codebook layer. This design avoids many of the challenges in the previous subsection. For example, a single point can only hold one bit of information, indicating the presence or absence of some feature, avoiding the challenges of holding multiple features and graded interpretations. Similarly, because the weight of non-activated codes is zero, the network cannot smuggle information through them.

However, there are several reasonable concerns one might have about features-as-points:

**Multiple codes per feature**  First, the network could hypothetically encode more complex features via complicated combinations of codes instead of assigning one feature to each code. For example, codes 1 and 2 together might represent happiness, while codes 1 and 3 together might represent cars. However, the simplicity of how the codes are chosen (by cosine similarity) makes it challenging to select codes with much complexity. Furthermore, similar concerns present themselves for continuous dictionary learning approaches where complex features are encoded via combinations of directions.

**Multiple features per code**  Second, the reverse failure mode might present itself: the model might still encode multiple features per code. Indeed, we have discussed certain cases where this is true, for example, in Sections 3 and 4. While some of this may be improved by choosing a larger codebook size or enabling the number of active codes $k$ to vary based on the input and position, it is unclear whether these approaches will solve the problem. Of course, as noted above, features-as-directions approaches may also suffer these failure modes.

**Lack of gradedness**  Third, one might worry that features-as-points cannot express the graded, continuous nature of many real-world features, such as sentiment. We share this concern; however, we note that there are mechanisms for expressing gradedness with discrete codes. For example, the network might choose to activate multiple codes in a given position or nearby positions or allocate different codes to different levels of the gradation. Furthermore, the strong language modeling performance of the codebook models suggests that the model can accomplish its task well despite this discrete constraint.

### E.2 ADDITIONAL BENEFITS AND TRADEOFFS OF CODEBOOK FEATURES

We list two additional differences between codebook features and dictionary learning approaches:

**Modification of the original network**  Dictionary learning approaches are typically trained off of a frozen network. By contrast, in codebook features, the pretrained network is typically finetuned to achieve high performance on the task with the codebook bottleneck. This training means we are interpreting a new network rather than the original one. Furthermore, the performance of this network is often slightly lower than the pretrained network, which is another tradeoff.

**Improved Efficiency**  Because codebook features use hard sparsity, only one large matrix multiplication is necessary (to compute similarity scores with each element of the codebook). In contrast,

a second large matrix multiplication may be needed by some sparse autoencoder approaches to do a full weighted sum over all $C$ dictionary elements rather than over $k << C$ elements chosen from the codebook; though activations such as ReLU may mitigate this problem to some degree. Furthermore, as we show in Appendix D.4, hard sparsity enables us to use libraries such as FAISS to replace the first matrix multiplication as well, further increasing efficiency.

### E.3 Mechanistic Interpretability

Researchers have long attempted to extract concepts, rules, and algorithms from neural networks. For example, a line of work since the late 1980s attempted to extract rules and finite automata from neural networks, especially recurrent neural networks (RNNs) (Servan-Schreiber et al., 1988; Elman, 1990, see (Jacobsson, 2005) for a review). A core challenge noted in these works is that neural networks use distributed representations (Rumelhart et al., 1986; 1988; Thorpe, 1989). This form of representation enables networks to represent more concepts than hidden units, at the expense of each unit no longer being interpretable (Elman, 1990). Thus, individual hidden units may not correspond to interpretable concepts, and a holistic analysis of the entire vector may be necessary to extract such structures (Servan-Schreiber et al., 1988; Elman, 1990; Jacobsson, 2005).

Recent work has attempted to revitalize this goal for today's much more expressive networks, attempting to detect concepts (Alain & Bengio, 2016; Kim et al., 2018; Olah et al., 2018; Goh et al., 2021; Bau et al., 2020b) and algorithms (Giulianelli et al., 2018; Clark et al., 2019; Olah et al., 2020; Bau et al., 2020a; Geiger et al., 2021; Geva et al., 2021; Elhage et al., 2021; Olsson et al., 2022; Wang et al., 2022; Chan et al., 2022; Friedman et al., 2023) inside of models, with many works focusing specifically on the challenges of neurons that fire on multiple concepts (Fong & Vedaldi, 2018; Olah et al., 2020; Mu & Andreas, 2020; Elhage et al., 2022b; Geiger et al., 2023), sometimes termed *superposition* (Olah et al., 2020).

Our work shares similar goals with the above works. Codebook features attempt to make identifying concepts and algorithms more manageable inside networks by refactoring their internal representations into a sparse and discrete form that is easier to understand and manipulate. We also discover one instance in Section 3 where codebooks represent more features than there are neurons, circumventing the superposition problem.

### E.4 Introducing Discrete Structure into Neural Networks

A range of works attempts to introduce discrete bottlenecks or structures into neural networks (Makhzani & Frey, 2015; Andreas et al., 2016; Keshari et al., 2019; Buch et al., 2021; Mao et al., 2019; Liu et al., 2023). Most saliently, vector quantization (Gray, 1984, VQ) is a classical technique in signal processing that was applied most prominently in machine learning through VQ-VAE (van den Oord et al., 2017) for use in autoencoder networks. By contrast, our method applies vector quantization to each hidden layer of any neural network (including autoregressive language models), enabling better understanding and control of the network's intermediate computation. Our grouped codebook method additionally employs product quantization (Jegou et al., 2010), an extension of vector quantization to multiple codebooks whose outputs are concatenated. Finally, our $k > 1$ models leverage ideas very similar to composite quantization (Zhang et al., 2014), where vectors from multiple codebooks are aggregated to represent the network; in our setting, it is the top-k vectors of the same codebook which are aggregated.

Another line of work introduces structured bottlenecks into training for interpretability and control. For example, concept bottlenecks (Koh et al., 2020) directly supervise an intermediate state of the network to align to a set of known features, while post-hoc concept bottlenecks (Yuksekgonul et al., 2022) enable transferring known features from another source (e.g., a multimodal model). In contrast to these methods, the concepts learned by the codebook are discovered *emergently* by the network as part of the training process. Another related work, Backpack Language Models (Hewitt et al., 2023), generate predictions by computing a set of weights over previous tokens; the next token is then predicted through a weighted sum of learned *sense vectors* associated with those tokens. By contrast, codebook features are applied to the *hidden states* of a neural network and facilitate better understanding and control of this via a sparse, discrete representation.

Work in computer vision has also explored vector quantization for image generation (Esser et al., 2021) and classification (Zhang et al., 2023), suggesting promising avenues for multimodal applications of these techniques.

### E.5 Editing or steering neural networks

Various methods attempt to control, edit, or steer the behavior of trained neural networks. A natural approach is to *finetune* the network on labeled data (Sermanet et al., 2013), though this process can be time- and resource-intensive and may distort the model's other capabilities. *Prompting* a model with natural language instructions (Brown et al., 2020) or control tokens (Keskar et al., 2019) is a lightweight steering method that overcomes some of these difficulties; however, not all models are promptable, and there may be instances where prompting is insufficient to ensure the model performs the desired behavior. In addition, a stream of work focusing on *model editing* makes targeted edits to concepts or decision rules inside of neural networks with a small number of examples (Bau et al., 2020a; Santurkar et al., 2021; Mitchell et al., 2021; Meng et al., 2022a;b).

Most related to our work, several recent works perform post-hoc steering of networks in ways that do not require per-edit optimization (Merullo et al., 2023; Hernandez et al., 2023; Turner et al., 2023) by adding vectors of different magnitudes to different layers in the network. Our work attempts to support the aims of such work by producing a sparse, discrete, hidden representation inside of networks. This representation makes it easier to localize behaviors inside the network (so that the user does not have to exhaustively perform interventions at every layer of the network to find the most effective intervention site) and makes it easier to perform the intervention by substituting codes (so the user does not have to try many different magnitudes of a given steering vector at each layer).

## F  Extended discussion of applications, significance, and future directions

### F.1  Uses for codebook features

While we primarily explore codebook features on transformer language models, our method is modality agnostic and can be applied to neural networks trained on any combination of modalities. We envision several different use cases for codebook features in such diverse contexts:

**Identifying phenomena in complex data**  Codebook features is an unsupervised method for discovering different latent features inside models. This method could be useful in situations where brainstorming novel kinds of features in data may be helpful for research. For example, codebook features could potentially help uncover new protein, genomic, or medical imaging data features by observing token activations and seeing what the examples all have in common.

**Feature detection**  In many applications, it is helpful to count the number of times a particular feature occurs or raise an alert when it does. While it may be more effective in many cases to collect a labeled dataset and train a classifier for a particular feature, codebook features are ready-made for this task and may enable faster iteration and experimentation.

**Counterfactual explanations**  One way of explaining a model's decision is via a counterfactual: would the model's decision change if this feature changed? While these counterfactuals often occur at the input level, codebooks enable counterfactual explanations at the hidden feature level.

**Steering models**  Finally, as explored in Sections 3.2 and 4.1, codebook features can be used to steer the complex generations of models. We anticipate the flexibility of this method to improve as codebook features are better understood.

### F.2  What this says about transformer computation

As seen in Table 5, codebooks enforce a strong information bottleneck between layers. We find it surprising that neural networks can operate amidst such a strong information constraint; this suggests that the underlying computation happening inside these networks is or can be made sparse along a set of understandable features.

### F.3   FUTURE WORK

We see several exciting directions for future work:

**Understanding circuits and weights**   Past work has investigated *circuits* in vision models, where more complex features are built up out of smaller features (see Appendix E for a full overview). The sparse and discrete nature of codebooks may make it far easier to identify such circuits, including in language models, due to the smaller number of possible relationships between components across layers. The discrete nature of codebooks also makes it easier to compute which codes tend to fire together across layers without the added complexity of accounting for continuous-valued neurons or feature directions. Understanding the relationship between activations across a single layer may also enable a better understanding of the *weights* of that layer, as these determine the input-output relationship the layer must produce.

**Understanding adversarial examples**   In computer vision, adversarial examples are small perturbations added to images that cause the network to misclassify them; for example, misclassifying a cat as a dog (Goodfellow et al., 2014). Codebooks enable identifying which codes in the network shifted to produce that change in decision: for example, was a cat ear feature changed to a dog ear feature? The discrete nature of codebook activations may also enable better defenses against adversarial attacks.

**Improving interpretability in larger models**   While we found that single-layer codebook models produced codebooks where the majority of codes had a comprehensible interpretation, in larger models, there were many codes where this was not the case. Future work might consider training models with even larger codebooks to capture the greater number of features the models represent. Future work might also consider using co-occurrence statistics of code activations to investigate whether there are codes that routinely fire together and may represent a single feature in tandem.

**Better quantization methods**   While we explore a simple cosine similarity–based approach in our paper, other methods for sparse quantization of activations (e.g. recent variational sparse coding methods (Tonolini et al., 2020; Fallah & Rozell, 2022)) may yield further gains.

**Understand shared representations across domains and modalities**   Recent work has shown generalization across distributions: for example, multimodal models contain neurons that fire on concepts (e.g., spiderman) in both text and image form (Goh et al., 2021), and language models trained on multiple languages can generalize zero-shot from one language to another (Johnson et al., 2017). Codebooks may enable tracing exactly how and where these features are integrated across the network.

