# OpenReview forum: "Codebook Features: Sparse and Discrete Interpretability for Neural Networks"
_ICLR.cc/2024/Conference — Submitted to ICLR 2024_

### Official Review · Reviewer_v2JW · 2023-10-27

**Soundness:** 3 good
**Presentation:** 3 good
**Contribution:** 3 good
**Rating:** 5
**Confidence:** 3

**Summary:**

The paper explores the concept of "codebook features" to make the hidden states of neural networks sparse, discrete, and more interpretable. By introducing a vector quantization bottleneck at each layer of the network, the authors achieve this sparse and discrete representation with only a modest performance degradation. The resulting codebook features serve as a promising unit for understanding and controlling neural network behavior, validated through experiments on finite state machines and large-scale language models.

**Strengths:**

* The authors show a method of using codebook features introduces a novel way to create sparse and discrete hidden states in neural networks. This approach facilitates the unsupervised discovery of both algorithmic and linguistic features within language models, tackling challenges like the superposition problem and thereby advancing the field of interpretability.

* The paper successfully demonstrates that the sparse, discrete nature of codebook features simplifies the complexity of a neural network's hidden state. This makes it easier to identify specific features and control network behavior, suggesting that this could be a powerful tool for more granular and sophisticated control in future applications.

**Weaknesses:**

* While Transformers are prevalent, there are many architecture differences between different models (e.g. novel layers, group-query attention, etc.). In this sense the study is limited in scope by focusing only on Transformer neural networks and examining their performance on a singular algorithmic dataset and two natural language datasets. This leaves unanswered questions about the generalizability of codebook features to other neural network architectures or different types of data, such as visual information.

* While the paper demonstrates the capability of codebook features in topic manipulation for language models, it does not explore other linguistic features like sentiment, style, or logical flow. This limitation narrows the understanding of how versatile and broadly applicable codebook features might be for controlling various aspects of neural network behavior.

**Questions:**

1. In authors' two-phase method for understanding and controlling the network's behavior, you focus on generating hypotheses for the role of codes and then steering the network by activating these codes. How robust is this method to the presence of adversarial or noisy input?

2. Authors mention that codebook features reduce the complexity of a neural network’s hidden state, making it easier to control the network’s behavior. Could you provide more details on the trade-offs involved? Specifically, how does the use of codebooks affect the model's capacity for generalization across different tasks or data distributions?

---

> ### Author Response · Authors · 2023-11-17
> **Response**
>
> We thank the reviewer for their review! We are glad they thought highly of the soundness, presentation, and contribution of the work.
>
> **Generalizability to other forms of neural networks**
>
> It is certainly true that we focus on Transformer neural networks in this work due to how widely used the architecture is. We present results applying codebook features to different parts of the transformer (attention and MLP layers), different size transformers (from 1 to 24 layers) and different datasets (synthetic data to natural language). We agree that an exciting direction for future work is in extending our method to other modalities and network architectures.
>
> **Other ways to control the network such as sentiment, style, and logical flow**
>
> We certainly agree; as we note in our paper, this is an exciting opportunity for future work.
>
> **How robust is the method of generating hypotheses and then steering the network?**
>
> In our experiments, we were able to find topic codes across a wide range of inputs, including a wide range of different domains within Wikipedia as well as the TinyStories dataset. However we agree that studying the relationship between codebooks and adversarial robustness would be interesting more broadly; we discuss this direction in a bit more detail in Appendix G3.
>
> **Tradeoffs of codebook features and generalization to other data distributions**
>
> Yes, there are a few trade-offs to using codebooks. First, codebooks require additional computations to be performed on the network in exchange for their interpretability benefits. Second, the performance of the codebook model is often slightly lower than the original model, although the difference is small in practice (Table 2). In general, the loss of a language model is a good measure of its performance across distributions it is evaluated on, so the small difference in loss between the codebook and base models aligns with our experience that the model continues to perform well across distributions.

---

### Official Review · Reviewer_aakk · 2023-10-31

**Soundness:** 3 good
**Presentation:** 3 good
**Contribution:** 2 fair
**Rating:** 6
**Confidence:** 3

**Summary:**

This paper proposes to select Top-k hidden units with the highest similarity score to learn sparse and discrete codebook features in an unsupervised way. They have also tried to apply this technique to transformers for language modeling tasks. Experiments show that the model can do well in the task of topic manipulation.

**Strengths:**

The paper is well-written and easy to follow.  The proposed method is straight forward. The experiment results are interesting in table 4. I believe that the proposed method may be applicable to many use cases, which potentially can lead to applications in future work.

**Weaknesses:**

1. The paper is missing the comparison for computational time. With a sparse and discrete codebook, it should lead to increased efficiency.
2. A chart or figure showing the learned topics for each layer may be missing.

**Questions:**

1. How do you find the activated codes and their corresponding topics? How do you know for example 'code 123' leads to the topic of 'dragon'?
2. Will the model lead to better computational time?
3. There are dead codes during training and how do we avoid them? Will the activated codes learn repeated semantics (For example, will one of the codes be activated all the time)?

---

> ### Author Response · Authors · 2023-11-17
> **Response**
>
> We thank the reviewer for their review! We are glad they find the results interesting and applicable in a wide range of use cases.
>
> **Computational efficiency and sparsity**
>
> This is a good question. We have provided some information about computational efficiency in Table 9. Indeed, leveraging sparsity through fast maximum inner product search does increase the throughput of the model. We suspect that custom sparse kernels would continue to increase the efficiency of codebook features further.
>
> **How do we associate codes with topics?**
>
> We follow a two-stage process. First, we take the code and retrieve examples where that code is activated. A person (or in the future, a language model) then looks at the examples and develops a hypothesis for the role of the code (e.g. “dragon,” if many dragon-related passages are retrieved). We then verify this role by activating the code during generation and seeing if the model’s behavior changes as we expected. Section 2.2 describes this process in more detail, and Sections 3 and 4 describe our automated evaluations for verifying the role of codes.
>
> **How are dead codes avoided?**
>
> While we see some number of dead codes during training, they do not significantly impact the success of the method. For example, in the 410M parameter model we train, fewer than 2% of codes were dead across the 384 codebooks we train on all the attention heads. Moreover, these dead codes do not appear to negatively affect the performance of the model, and they can be discarded after training.
>
> **Are some codes activated all the time?**
>
> We do see a small number of codes that are activated very frequently (see Figure 7). However, the majority of codes are activated far more rarely. The codes that occur most frequently tend to be associated with very general topics or very common words (e.g. articles like “the”).

---

> > ### Comment · Reviewer_aakk · 2023-11-22
> >
> > Thanks for the authors' reply. I think most of my concerns have been addressed. I think it would be better if the author could explain more about the motivation for why we need to use sparse and discrete codes in the main paper. Could we just use discrete codes or just sparse codes? Why do we need to have both discrete and sparse? I will keep my score as it is.

---

> > > ### Author Response · Authors · 2023-11-22
> > > **Thank you**
> > >
> > > Thanks to the reviewer for engaging with our response! We are glad to hear that most of their concerns have been addressed.
> > >
> > > We agree that the question of "why both sparse and discrete?" is a useful framing, and will include more discussion in our next revision. Briefly: "Sparse but not discrete" corresponds to dictionary learning, and we will link to our discussion of the benefits of codebooks over dictionary learning approaches in Appendix E.1. "Discrete but not sparse" could suggest a very large number of active states, which would make it overwhelming to interpret the hidden states of the model. This relates to our analysis of "codebooks as information bottlenecks" in Appendix B, which we will link to as well.
> > >
> > > Thank you again for your time, and we are happy to continue engaging with any follow up questions!

---

### Official Review · Reviewer_AYvz · 2023-11-01

**Soundness:** 2 fair
**Presentation:** 3 good
**Contribution:** 2 fair
**Rating:** 5
**Confidence:** 3

**Summary:**

In this work, a method is proposed to improve the interpretability and controllability of a transformer network by quantizing the activations per token with a sparse combination of entries from a codebook. To select the sparse codes, the cosine similarity between token activations and codebook entries are computed and a weighting is taken based on top-k most similar entries. Experiments on a dataset of state transitions and on language datasets show that the use of codebook entries correlate with certain aspects of the dataset (e.g., states or semantic concepts). Furthermore, experiments show that codebook entries can be applied as a token to steer the output of the transformer network.

**Strengths:**

* This work considers the problem of interpreting the intermediary layers and controlling the output of transformer networks, which are of significant interest to the machine learning community. Furthermore, quantizing features with codebooks is a popular technique which the work demonstrates leads to a minimal degradation in model performance.
* Experiments with the TokFSM dataset are intuitive and clearly demonstrate the ability to intervene in the output of a transformer trained with codebook quantization. Experiments such as the JS divergence with the target token distribution in Figure 4, are used to demonstrate effective intervention. Given that one has access to the semanticity of entries in the learned codebook, the proposed method seems to be effective at steering the output of transformer network.
* The work contains a comprehensive related works section in the appendix that clarifies the benefits of a discrete codebook over using a dictionary approach (referred to as “features-as-directions”)

**Weaknesses:**

* One shortcoming of the experiment in table 1 is that it has two dependent variables: both the quantization level k and the codebook size C. It is important to ablate changes to these two variables separately to better understand if they both independently provide benefits. There is a similar issue of two dependent variables being tested at the same time in Table 2 where both k is modified as well the features being quantized (i.e., attention vs mlp features).
* The benefit of sparsity in the combination of codebook entries has not been clearly articulated. In fact, in section F.1.1 of the manuscript, it is argued that the continuous combination of different atoms reduces interpretability. Selecting codebook entries via top-k cosine similarity is a naive approach that has not been compared to more recent sparse coding techniques (e.g., variational sparse coding methods like in Tonolini et al. 2020 and Fallah et al. 2022). Except for the potential increase in modeling capacity, the work does not demonstrate the benefit of quantizing with multiple codebook entries per token.
* Interpretation of codebook entries still seems to require manual intervention. It requires a user to find input data for which a codebook entry is often activated, which can be timely and costly. It is unclear how one would use current methods to find a codebook entry that corresponds with a certain semantic concept without manually performing a forward pass using data corresponding to that concept.


Minor:
* Some citations need revising in the bibliography (e.g., “On the role of scientific thought”).

**Questions:**

* Can the authors clarify the benefit from increasing k? Since the authors use cosine similarity between the activations and the codebook to pick the top-k entries, what would the difference be between each of these k codebook entries? It seems that taking k entries contradicts the viewpoint of “features-as-points”, and may even lead to what the authors refer to as “smuggling of information”. I would expect that increasing k may improve performance of the model, at potential cost to interpretability.
* Out of curiosity, have the authors considered quantizing each vector along the feature components (i.e., divide the N features of each token into k blocks)? Is this what the authors refer to as “grouped codebooks” in E3? If so, I believe this warrants discussion and attention in the main text. This would be an alternative to quantizing each token with k codebook entries that are very close in cosine similarity and be closer to the VQ-VAE setting.
* The notation and presentation of the different loss terms in section 2.1 can be made more clear. In the cross-entropy loss, x is used to denote a categorical random variable corresponding to a token being selected. In the reconstruction loss, to my understanding, x is referring to a continuous random variable corresponding to the activation in an intermediary layer, even though the variable a is used in an earlier section. Furthermore, k-codes are combined to quantize each activation. Is the MSE taken with the sum of these codes or each code individually?

---

> ### Author Response · Authors · 2023-11-17
> **Response**
>
> We thank the reviewer for their review!
>
> **Untangling dependent variables in Table 1**
>
> This is a good point. To address this issue, we reran our experiments to hold $C$ constant while changing $k$. We include these results in Table 1, which mirror the results from our previous version. We also include an experiment with an MLP codebook with $k=8$ and $16$ groups in Table 2; this enables a direct comparison to the attention $k=8$ model because our model has $16$ attention heads.
>
> **Reason for using multiple codes per layer**
>
> Yes, the primary benefit of working with $k>1$ codebooks is expressivity; because models likely represent more than 1 feature per layer, our interpretability method needs to be able to model multiple features. While we chose a simple method for our initial experiments, we agree that more sophisticated methods, including variational sparse coding approaches, are promising. We have mentioned this, including Tonolini et al. 2020 and Fallah et al. 2022, in our newest revision.
>
> **What happens with increasing k?**
>
> Yes, increasing $k$ does improve the performance of the codebooks model. In our updated Table 1, we show several runs where we only change $k$, demonstrating steady improvements in performance. In terms of interpretability, we agree that in the limit increasing $k$ could make the hidden state harder to understand. However, as Table 5 shows, even large values of $k$ transmit many fewer bits of information than a standard neural network layer, limiting the complexity of the hidden state. We will include this discussion in our revision.
>
> **Additional motivation for top-k, and whether the top-k codes have different roles**
>
> High dimensional spaces enable code vectors containing different information to be selected through top-k cosine similarity. As an example, consider the vector [$a$; $b$], the concatenation of two smaller vectors $a$ and $b$. This vector would have relatively high cosine similarity with [$a$; 0] and [0; $b$], despite the two vectors being potentially very distant from each other. Thus, we believe the $k>1$ models are fully compatible with the features-as-points viewpoint. However, there are other cases we have seen where several of the $k$ codes appear to activate on similar tokens. This may indicate that the model needs to represent fewer features than $k$, or that the redundancy is helpful for the network computation.
>
> **Is interpretation manual?**
>
> Yes, currently interpreting the codebook features requires manual interpretation. However, other work (e.g. [1, 2]) has demonstrated that language models can be used to interpret neurons or features as well, which we expect to be a more scalable solution.
>
> **Does interpretation require forward-passes?**
>
> Yes, currently interpretation does require a forward pass over a subset of the data. However, forward passes on this subset of data can be computed once and then the resulting data can be used to interpret the entire network. In practice, we expect the amount of compute needed to do this will be much smaller than that used to finetune the network to produce the codebook features.
>
> **Grouped codebooks**
>
> Yes, we believe the reviewer’s description is the same as our grouped codebooks method. We have included more discussion of this in the paper, however we note (per the example given in the previous paragraph) the $k>1$ codebooks can implement something quite similar to grouped codebooks. Thus, it is possible the primary benefit of grouped codebooks may be in improving optimization of the network, rather than in the expressivity or interpretability of the codebook itself.
>
> **Notation in 2.1**
>
> Yes, for the reconstruction loss, $x$ refers to the continuous activations. We changed this to $a$ in the most recent revision. The MSE is taken with respect to the output of the codebook (i.e., the sum of the codes).
>
> [1] Natural Language Descriptions of Deep Visual Features, Hernandez et al 2022
>
> [2] Language models can explain neurons in language models, Bills et al 2023

---

> > ### Comment · Reviewer_AYvz · 2023-11-19
> > **Response to Rebuttal**
> >
> > Thank you for the clarifications in your response to the review. It is still unclear to me how to reconcile the use of `k > 1` with the "features-as-points" perspective. You provided one possible way this may occur, but is there any evidence that suggests that the `k` different codes represent different features? I view this as an important point, since the possibility of the network learning something closer to "features-as-directions" makes the token selection strategy (and potentially the weighting of the discrete tokens) very important. Also, the work currently contrasts codebook features with dictionary learning by stating the former prevents "smuggling of information." This seems to contradict the fact that with `k > 1` this phenomena would likely occur.

---

> > > ### Author Response · Authors · 2023-11-22
> > > **Response**
> > >
> > > Thank you for engaging with our response!
> > >
> > > **Empirical evidence that active codes can have two distinct features**
> > >
> > > Here is one example of how multiple codes that are activated can have distinct roles. Consider the input: *There was a bright pink dragon in the sky.* We examine the 8 active codes at the different heads of our TinyStories 1L language model at the token “dragon”:
> > >
> > > For the codebook at *head 4*, we see several types of distinct codes including:
> > > - "Dangerous entities": This code activates on dangerous animals and entities like dragon, bear, lion, tiger, dinosaur, snake, wolf, shark, monster, witch, etc.
> > > - "Flying entities": This code activates on animals and entities that can fly like dragon, bird, owl, crow, eagle, hawk, unicorn, ghosts, helicopter, etc.
> > > - "Dragon": This code activates on various instances of the word “dragon”.
> > >
> > > For the token at *head 6*, we see a different set of distinct codes, including:
> > > - "Previous token adjective": This code activates on a token when the previous token was some adjective (including colors) like yellow, red, blue, round, shiny, colorful, special, etc.
> > > - "Previous token color": This code activates on a token when the previous token was a color like blue, red, green, brown, etc.
> > > - "Animal followed by an adjective": This code activates when an animal is mentioned with an adjective like playful rabbit, furry cat, brown dog, little fish, etc.
> > > - "Previous token pink or green": This code activates on a token when the previous token was pink or green.
> > >
> > > Thus, we see that codes can capture different features relevant to a single token that are useful for the language modeling task, even with $k>1$. We will include these examples in our next revision and thank the reviewer for the suggestion.
> > >
> > > **Smuggling of information**
> > >
> > > At least as used in the SoLU paper [1], “smuggling of information” refers to how a continuous feature direction can carry information about one feature (e.g. “dragon”) at large magnitudes, giving the illusion of interpretability, yet at smaller magnitudes can carry unrelated information about other features. In codebook features, each individual code is discrete, meaning that if it has an understandable interpretation, there are no “smaller magnitude” activations for that code that could carry different information. Thus, the network cannot smuggle information in the same way.
> > >
> > > Both dictionary learning and codebook features are similar in that they allow for more than one feature (e.g. a point or a direction) to be active for any one activation vector ($k>1$ in our terminology). Thus, both have the issue that these features may or may not be interpretable and if they are interpretable may or may not represent different concepts. These are valid potential limitations that we attempt to discuss in Appendix E.1, but we believe this is distinct from the notion of “smuggling information.” We are happy to add this discussion to the next revision of our paper if it would improve the clarity of the manuscript.
> > >
> > > Thanks again for engaging and we are happy to answer any further questions.
> > >
> > > [1] Elhage et al, 2022, https://transformer-circuits.pub/2022/solu/index.html

---

### Official Review · Reviewer_XCvz · 2023-11-01

**Soundness:** 4 excellent
**Presentation:** 4 excellent
**Contribution:** 3 good
**Rating:** 6
**Confidence:** 3

**Summary:**

This paper proposes the discretization of intermediate features within deep Transformers, ensuring that the network's outputs (decisions) are contingent solely on finite, interpretable, and sparse codes. The authors demonstrate that by exploring the connections between specific codes and semantic or high-level topics and by adjusting these intermediate codes, users can exert intuitive control over the network's behavior. A comprehensive set of experiments reveals that the modified networks maintain competitive performance levels after fine-tuning.

**Strengths:**

* The authors introduce an intriguing inquiry into the performance of deep Transformers when their intermediate features are discretized. Surprisingly, the results appear promising for both small-scale and large-scale Transformers.
* Introducing discrete features (codes) which are shown associated to specific semantics in the paper brings interpretability to some extent. More importantly, such an approach enables users to control the models' output by modifying codes that have human understandable semantics. In fact, a similar approach has been introduced in computer vision to facilitate the creation of interpretable inference procedures [1] and controllable image synthesis [2].
* The paper is technically sound, and it provides a thorough discussion of the related literature.
* The paper is well-written and easy to follow.


>[1] Schema Inference for Interpretable Image Classification. (ICLR 2023)
>
>[2] Taming Transformers for High-Resolution Image Synthesis. (CVPR 2021)

**Weaknesses:**

[Major]
1. **Experiments:** As mentioned in the paper that a code can be related to some specific semantics; however, the results supporting such claim appears insufficient. The authors may consider conducting comparative analyses of the distribution disparities between code associated with similar and dissimilar semantics to substantiate this claim.
In addition, is it possible that a (some) certain code(s) may correspond to a multitude of distinct semantic contexts?
2. **Experiments:** It is interesting that model outputs is controlled by the codes. However, based on the results on TokFSM dataset, it appears that the code following the MLP layer plays a more significant role. Nevertheless, the experiments conducted by the authors on WikiText-103 dataset only involve discrete attention layers (in Table 2 (b)). Does this incongruity potentially render it challenging to control the model?
3.
4. **Experiments:** In Section 2, the authors sum top-k codes weighted by the same value (specifically, 1). How will each code influence to the model's decision? (For example, does codes having semantics related to the target task contributes most to the model's decision evaluated by attribution methods?) In particular, the authors can utilize feature attribution methods [3-5] to present quantitative and qualitative analyses.
5. What is the extent of the contributions made by these pieces of code to the final outcome? It may be worthwhile to investigate this using feature ablation techniques to discern whether the words crucial for the model's decision-making align with human intuition.

    >[3] Deep inside convolutional networks: Visualising image classification models and saliency maps.
    >
    >[4] Did the model understand the question? (ACL 2018)
    >
    >[5] Analyzing Chain-of-Thought Prompting in Large Language Models via Gradient-based Feature Attributions.

[Minor]
1. The font size in the figures is excessively small, making them particularly challenging to decipher when printed (e.g., Figure 1, 2 and 3). Furthermore, it is advisable for the authors to employ vector graphics to enhance the quality of the illustrations.
2. The authors do not provide codes for reproducibility check.
3. The authors could provide some failure cases to facilitate further analysis of how the proposed method yields incorrect results. If feasible, this could also serve as a basis for advancing future work.

**Questions:**

My questions are listed in the "Weaknesses" section. I am looking forward to the authors' relply.

---

> ### Author Response · Authors · 2023-11-17
> **Response**
>
> We thank the reviewer for their review! We are glad they found the results surprising and promising.
>
> **Can multiple concepts be associated with a single code?**
>
> It is certainly possible that a code could be associated with multiple distinct concepts. We study this in Appendix D.4 and Figure 8, where we find that almost every code holds a single concept at the first layer. In later layers, we find that more concepts share a code, however we observe a good reason for this—namely, that these concepts are related to one another and tend to share the same next token (Figure 9). We agree that further investigation of this question is a fruitful direction for future work and have noted this in our revision.
>
> **Why we use attention codes in one model and MLP codes in another**
>
> This is a good question. The reason we use different types of codes here is that we are trying to control different aspects of the sequence/text in each model. In the TokFSM environment, we are trying to alter the prediction of an individual state/token. We find codes in the MLP layers are most associated with these single tokens. For the language modeling experiments, we are trying to alter the global topic of a generation. Topics typically manifest across many tokens, rather than a single token, and we find the attention layers are most associated with these features. However, we believe it is likely that, e.g., for more local linguistic features (such as word choice) editing the MLP codes in a language model may prove to be the best choice.
>
> **”In Table 3, the authors have presented PriViT and MPCViT in distinct hyper-parameter configurations, giving rise to concerns regarding the fairness of the comparisons.”**
>
> We believe this comment may be about another paper, as we do not present or discuss results on methods called PriViT or MPCViT.
>
> **How does each code influence a model’s decisions?**
>
> This is a good question. To evaluate this, we choose several families of topic codes and compute how varying the number of topic codes we activate changes the fraction of generations that follow that topic. As expected, we find that increasing the number of topic codes changes the fraction of decisions (Figure 10). Qualitatively, when we look at the activated tokens, we note that there is significant overlap in the tokens activated by each code; however, there are still many tokens which are activated by some codes and not others, suggesting that the multiple codes together may help the network form a more robust version of the topic concept.
>
> **Computer vision papers**
>
> We thank the reviewer for the pointers to related work and have included them in our most recent revision.
>
> **Code for reproducibility**
>
> We have included an anonymized codebase in the newest supplementary materials of our paper. We have also uploaded the same codebase along with our model weights here: https://drive.proton.me/urls/C205JG7GQM#l7CfinlSZUHM
>
> **Vector graphics**
>
> The newest version of our paper now uses vector graphics for most of the figures, which should result in improved readability. We thank the reviewer for the suggestion.

---

> > ### Comment · Reviewer_XCvz · 2023-11-20
> >
> > Thanks for the authors' reply, and my main concers have been addressed. In addition, I hope the authors could add the discussion  about "Question 2" in the revised version. In spite of the interesting topic of the paper, I am not an expert in NLP, and I will keep my current score (6).

---

> > > ### Author Response · Authors · 2023-11-21
> > > **Thank you**
> > >
> > > We thank the reviewer for reading and engaging with our response! We are very happy to hear the reviewer's main concerns have been addressed. We have added an additional paragraph in Section D.8 with the discussion about Question 2. We are happy to continue engaging with the reviewer if they have any further questions or uncertainties (whether from an NLP or general machine learning perspective).

---

### Author Response · Authors · 2023-11-17
**General response**

We thank the reviewers for their comments and helpful suggestions! We have uploaded a revision of our paper with the new experiments and changes described in our responses below. We have also uploaded an anonymized codebase to the supplementary material to facilitate reproducibility. We appreciate the reviewers' time, and are happy to continue answering questions to address any lingering issues.

---

### Meta-Review · Area_Chair_Mp93 · 2023-12-14

**Metareview:**

The work proposes to incorporate a layer of sparse and discrete codes into a text language model or a text classifier. The demonstrated benefit of the discrete code is the ability to change the topic of the generated text in language models.
The paper explores incorporating this codebook layer into Transformers and in the text domain (while leaving CNNs and images for future work).

In response to the reviewers (`aakk`, `AYvz` ), the authors have sufficiently discussed the difference between this work (learning sparse and discrete codes) and related approaches of Dictionary Learning (learning sparse but not discrete code) and learning discrete but not sparse codes.

While the overall sentiment is that the work direction is interesting and meaningful, the ratings stay at borderline (5,5,6,6).
According to AC, the main weakness is the lack of a demonstration of the **utility** of the sparse and discrete codes learned (point #5 in the review of `XCvz`).
For example, if the authors state that these codes are "interpretable", then a human study should be run to collect the evidence of how such interpretability provides benefits to a downstream task.
If the goal is to improve the controllability of the generated sentences, then multiple text-generation properties need to be tested e.g. the style, flow, and sentiment of the generated text (raised by reviewer `XCvz`). However, only `topic` controllability is tested.

Overall, the AC finds the direction of this work very interesting and meaningful. Yet, for publication, the current version of the work needs more experiments to demonstrate the usefulness and utility of the sparse and discrete codes to justify the main goal of `interpretability` as written in the title.

**Justification For Why Not Higher Score:**

The work lacks a solid demonstration of the usefulness of the interpretability on a downstream task.

**Justification For Why Not Lower Score:**

N/A

---

### Decision · Program_Chairs · 2024-01-16

Reject